# Neuromorphic antennal sensory system

Chengpeng Jiang [1,2], Honghuan Xu[1,2], Lu Yang[1,2], Jiaqi Liu[1,2], Yue Li[1,2], Kuniharu Takei [3] ✉ & Wentao Xu [1,2] ✉

Insect antennae facilitate the nuanced detection of vibrations and deflections, and the non-contact perception of magnetic or chemical stimuli, capabilities not found in mammalian skin. Here, we report a neuromorphic antennal sensory system that emulates the structural, functional, and neuronal characteristics of ant antennae. Our system comprises electronic antennae sensor with three-dimensional flexible structures that detects tactile and magnetic stimuli. The integration of artificial synaptic devices adsorbed with solution-processable $MoS_2$ nanoflakes enables synaptic processing of sensory information. By emulating the architecture of receptor-neuron pathway, our system realizes hardware-level, spatiotemporal perception of tactile contact, surface pattern, and magnetic field (detection limits: 1.3 mN, 50 μm, 9.4 mT). Vibrotactile-perception tasks involving profile and texture classifications were accomplished with high accuracy (> 90%), surpassing human performance in "blind" tactile explorations. Magneto-perception tasks including magnetic navigation and touchless interaction were successfully completed. Our work represents a milestone for neuromorphic sensory systems and biomimetic perceptual intelligence.

Biological tactile sensory organs, encompassing skin, whiskers, antennae, among others, have manifested in diverse forms, each possessing unique anatomical structures, sensory functions, and neuronal encoding or processing mechanisms. Mainly, these organs demonstrate proficiency in mechano-sensation related to pressure and vibration, utilizing spatiotemporal encoding methods to interpret somatosensory information obtained from various mechanoreceptors. These intricate systems endow a refined perception of textures, profiles, and shapes during both tactile interaction and active exploration[1–3]. Inspired by nature, artificial tactile sensory systems, including e-skin and e-whisker, have been reported, contributing to the ongoing development of skin electronics and epidermal electronics[4–7]. State-of-the-art artificial mechanoreceptors further combine neuromorphic devices/circuits and biomimetic tactile sensors to impart processing and memory functions[8–11]. However, previous efforts to emulate natural tactile sensory organs and nervous systems mainly focused on the planar, multilayer design of sensor (e-skin) and the multi-directional sensation of force and strain (e-whisker)[6,7,12–15]. These

systems mostly rely on mimicking the skin and hair in mammals, imposing limits on their structures and functions.

Insects' tactile sensory organs, despite their diminutive scale and paucity of neurons compared to mammals, exhibit efficient processing and multimodal sensory functions encompassing mechano-perception, magneto-perception, audio-perception, and chemo-perception[2,16,17]. These capabilities could serve as blueprints for the development of biomimetic sensory platforms. Hallmarked by their segmented, flexible, three-dimensional architecture, insect antennae deliver exceptional mechano-sensory performance in response to deflections and vibrations[18]. Some research grounded in cellular and molecular evidence tentatively hypothesizes that specific insects, such as ants, enable magneto-reception via their antennae's mechanically sensitive, magnetite-infused magnetoreceptors[19–27]. The exquisite antennal structures, densely innervated with sensory receptors and neurons, emit spatiotemporally-encoded neural spike sequences, allowing for the detection of vibrotactile and magnetic stimuli. The perceptual acuity is comparable to, or even surpasses, that of human

[1]Institute of Photoelectronic Thin Film Devices and Technology, Key Laboratory of Photoelectronic Thin Film Devices and Technology of Tianjin, College of Electronic Information and Optical Engineering, Engineering Research Center of Thin Film Photoelectronic Technology of Ministry of Education, Smart Sensing Interdisciplinary Science Center, Nankai University, Tianjin, China. [2]Shenzhen Research Institute of Nankai University, Shenzhen, China. [3]Graduate School of Information Science and Technology, Hokkaido University, Sapporo, Japan. ✉e-mail: takei@ist.hokudai.ac.jp; wentao@nankai.edu.cn

skin, thereby enabling insects to execute complex tasks, including foraging, object identification, and navigation[2,22,28,29]. Nevertheless, tactile sensory systems inspired by insect antennae (antennal sensillum as well) are yet to be fully explored. It is envisioned that artificial tactile sensory systems, mimicking the structural, functional, and neuronal characteristics of insect antennae, will enable multimodal perception in highly efficient and biologically plausible manners. The development of these insect-inspired systems has the potential to break the design constraints of skin electronics, leading to the realization of tactile intelligence and perceptual augmentation for advanced robotics and human-machine interfaces.

In this work, we report a neuromorphic antennal sensory system designed for vibrotactile- and magneto-perception. The electronic-antennae sensor in this system features biomimetic, flexible, three-dimensional (3D) structures. The sensor's responses to vertical compression, lateral scanning, and magnetic proximity are separately measured in different operation modes to assess its sensing capabilities for pressure, vibration, and magnetic stimuli. The artificial synaptic device in the system, designed with dual planar gates, is fabricated through the liquid-phase adsorption of two-dimensional (2D) nanoflakes of transition metal dichalcogenide (TMD) onto a metal oxide film. Subsequently, the device's performance in processing spatiotemporal spiking signals is investigated. This system adopts the connection architecture of receptors and neurons and also employs the encoding strategy of fast-adapting (FA; sensitive to dynamic stimulation) and slowly-adapting (SA; sensitive to static stimulation)

mechanoreceptors to imitate the neural pathway and neuronal coding observed in biological antennae. Neuromorphic tactile- and magneto-perception experiments, including chess profile classification, Braille code recognition, surface discrimination, magnetic material classification, magnetic navigation, and touchless interfacing, were conducted to validate the system's potential applications in tactile cognition, sensory robotics, and smart interfaces. Unlike the extensively studied "e-skin" system, our neuromorphic antennal sensory system ("electronic antennae") incorporates biologically plausible designs that mimic the insect antennae in terms of structure, function, and neural encoding/processing.

## Results

### Neuromorphic antennal sensory system

In a vast array of insects, encompassing flies, bees, and ants, antennae emerge as intricately complex, multifunctional sensory apparatuses. Antennae underpin various sensory modalities, including mechanoreception, magnetoreception, audio-perception, and chemoreception[2,17,22,30]. Examining the case of ants, these creatures exploit their paired antennae to achieve tactile and magneto-perception with notable acuity (Fig. 1a). The segmented structure of these antennae functions as a sensorimotor organ that can actively maneuver for mechanosensation of contact, vibration, and surface texture[16]. Furthermore, certain literature hypothesizes that the ion-rich particles present within the antennae potentially facilitate the ant's perception of the earth's magnetic field via the detection of the

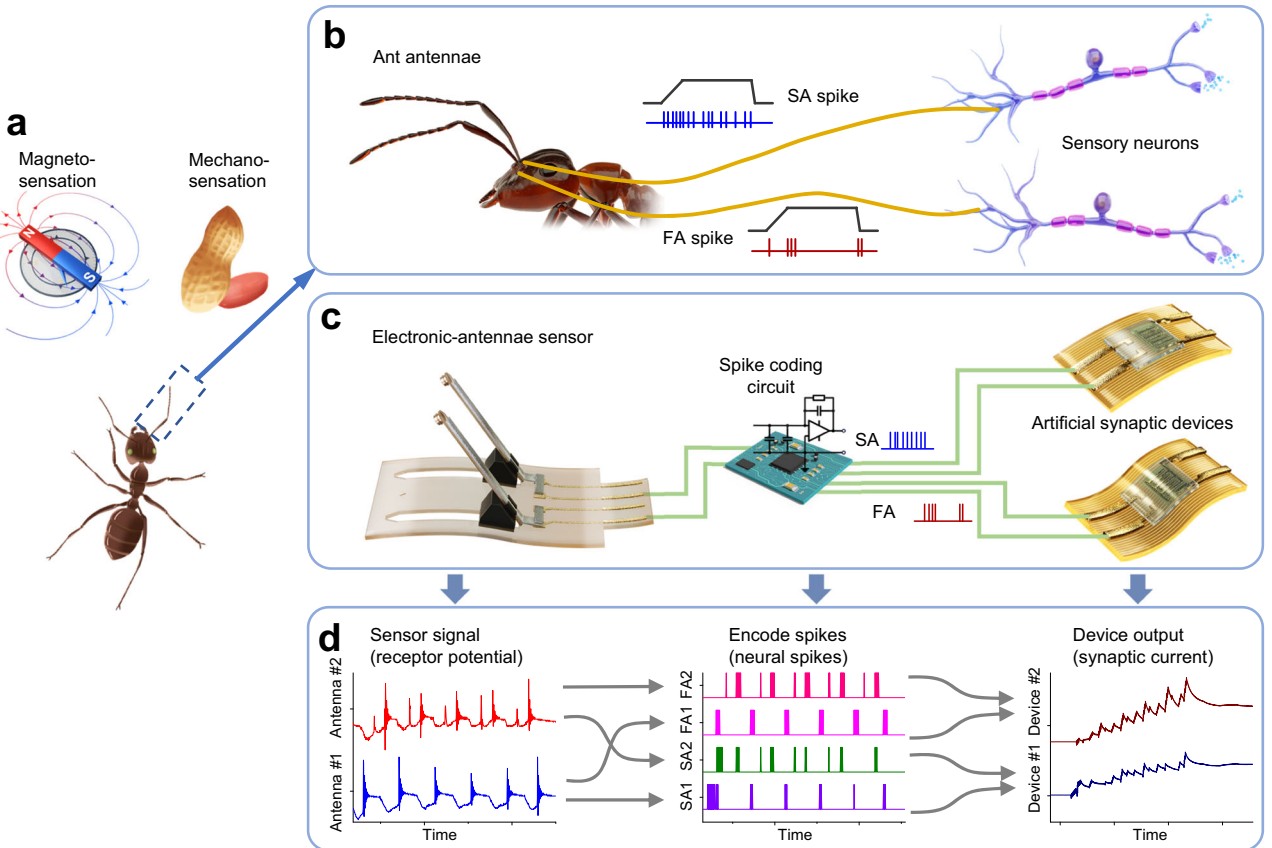

**Fig. 1 | Design of the neuromorphic antennal sensory system. a** Mechano- and magneto-sensation functions of the ant. **b** The architecture of a biological antennal nerve. Slowly-adapting (SA) and fast-adapting (FA) neural spikes are transmitted from sensory receptors to sensory neurons. **c** A neuromorphic antennal sensory system comprises an electronic antennae sensor, a spike-encoding circuit, and artificial synaptic devices. **d** Information flow in neuromorphic antennal sensory system. First, the piezoelectric signal (receptor potential) acquired from each

artificial antenna is encoded into SA and FA spike trains carrying spatiotemporal patterns of the sensory stimuli. Then, two artificial synaptic devices (SA and FA devices) process the pairwise SA and pairwise FA spike trains, respectively, and produce two synaptic currents. Curves in (**d**) are shifted vertically for clarity. SA1 and SA2: slowly adapting spikes from Antenna #1 and #2; FA1 and FA2: fast-adapting spikes from Antenna #1 and #2.

magnetic force applied on the antennae[19–27]. The antennae are replete with diverse sensory receptors and neurons, comprising what is known as Johnston's organ. These components serve to identify and transcode movements of the antennae into slowly-adapting (tonic firing for sustained stimuli) and fast-adapting (phasic firing for changing stimuli) neural spikes (Fig. 1b)[18]. This orchestration of spatiotemporal spikes, carrying sensory information, is subsequently processed in the central nervous system, facilitating multifunctional perception[3].

By emulating the structural, functional, and neuronal characteristics of ant antennae, a neuromorphic antennal sensory system was developed using an electronic antennae sensor, a spike-encoding circuit, and artificial synaptic devices (Fig. 1c, Fig. S1). The sensor, fabricated with a pair of magnet-loaded flexible artificial antennae made of PVDF and PET films, converts vibrotactile and magnetic stimuli into vibrations and deflections, generating piezoelectric signals resembling the "receptor potential" (Fig. 1d). The two piezoelectric signals acquired from the spatially separated artificial antennae were encoded into pairwise SA and pairwise FA spike trains with distinct temporal patterns, realizing spatiotemporal encoding. The two pairs of spike trains are transmitted separately to two artificial synaptic devices for

neuromorphic processing. The synaptic current of the devices and the mean firing rate of the pairwise spikes, both considered as the signature of sensory information, were obtained at the hardware level in an event-based manner (Supplementary Note 1), facilitating neuromorphic vibrotactile- and magneto-perception.

## Electronic-antennae sensor

A close examination of the ant's antenna divulges a multi-segment, elbowed structure that optimizes mechano-sensation (Fig. 2a). The binding of the short basal segment of the scape, characterized by its limited movement, and the elongated distal segment of funiculus, known for its unrestricted movement, embodies a pliant joint structure. This structure, innervated to the mechano-responsive sensory receptors, empowers both active tactile exploration and passive tactile sensation[2,18]. The design of our electronic-antenna sensor follows the working principle of ant antennae (Fig. 2b). A pair of antennae structures were created by bending two antennae patterns on a laser-cut plastic substrate using socket-like ramps (Fig. S2), which allow the top segment of the artificial antennae to deflect upon contact while keeping the bottom segment fixed. A piezoelectric film (metalized

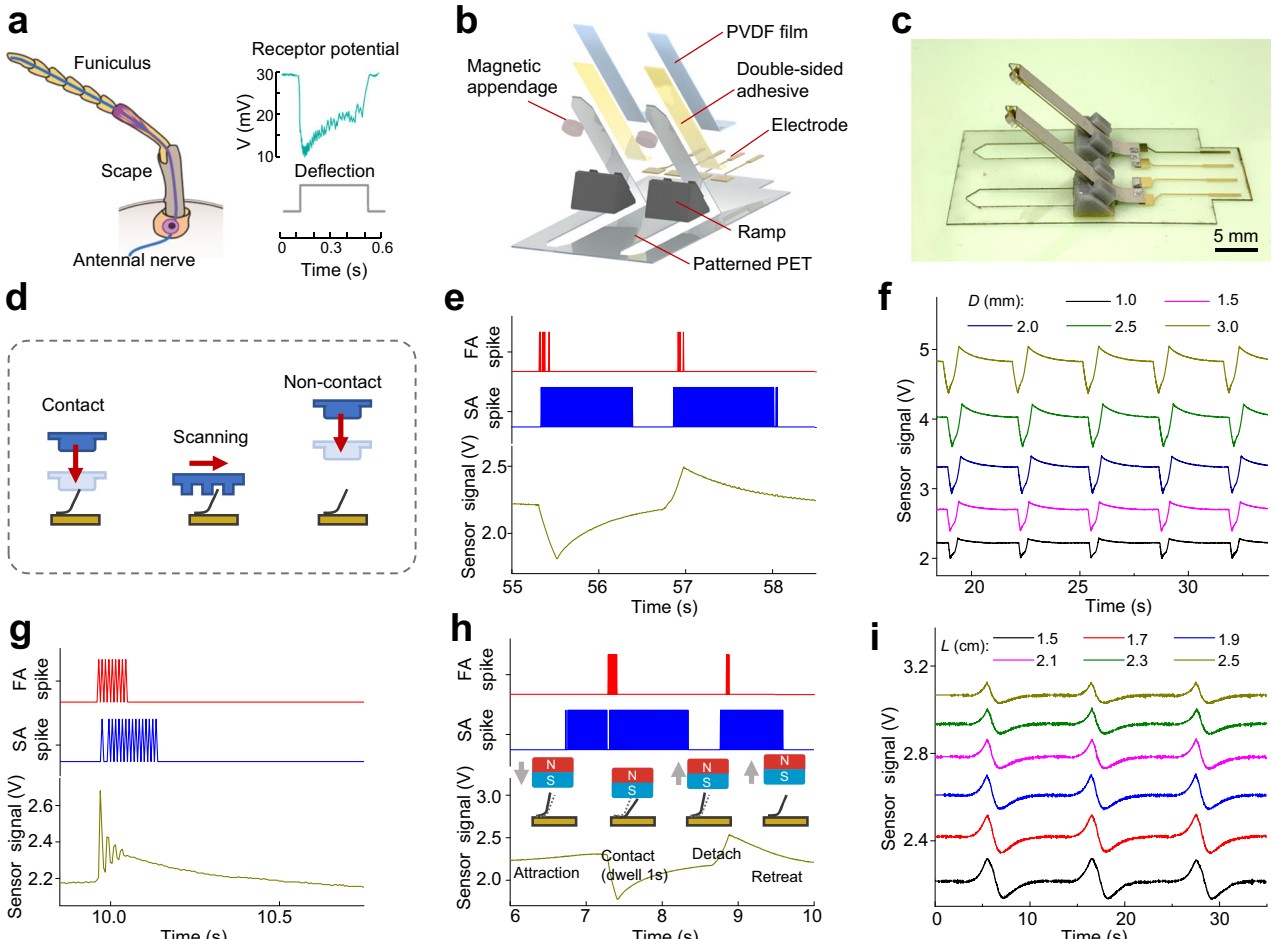

**Fig. 2 | Structure and performance of the electronic-antennae sensor.**
**a** Illustration of ant antennae composed of segments of scape and funiculus innervated by the antennal nerve. Sensory cells of the antennae exhibit a sudden decrease followed by a gradual increase of the receptor potential under static deflection. (Adapted from ref. 18 with permission from Elsevier).
**b, c** Exploded-view schematic illustration (**b**) and photograph (**c**) of the electronic-antennae sensor with tactile- and magneto-sensation functions.
**d** Illustration of the sensor's different operation modes (contact, scanning, non-contact modes) for perceiving touch, vibration, and magnetic proximity.
**e** Detection of static tactile stimuli in contact mode (dwell time 1 s). Time-

resolved sensor signal and corresponding spike trains (SA and FA) were recorded. **f** Detection of static tactile stimuli in contact mode (no dwell time) under different deflections (D) of the artificial antennae. **g** Detection of vibrotactile stimuli in scanning mode. **h** Detection of magnetic proximity in contact mode (dwell time 1 s). Deflections of the artificial antennae during the attraction, contact, detaching, and retreat process are schematically shown.
**i** Detection of magnetic proximity in non-contact mode (no dwell time) under different sensor-to-object distances (minimum value L). Curves in (**f, i**) are offset vertically for clarity. PVDF Polyvinylidene fluoride, PET Polyethylene terephthalate, SA Slowly adapting, FA Fast-adapting.

surface) was attached to the surface of the artificial antennae (Fig. S2). Tactile force-induced bending of the sensor generated a piezoelectric signal, providing tactile-sensation functionality (Supplementary Note 2). Additionally, a magnetic appendage (uniaxially magnetized) was installed on the tip of the artificial antennae (Fig. S2). The presence of a magnetic or ferromagnetic object induced deformation of the artificial antennae through magnetic interaction, thereby enabling magneto-sensation functionality (Supplementary Note 2). The thickness of the plastic substrate was optimized (180 μm), as it is related to the spring constant of the antennae structure (power-law relationship) modeled as a cantilever beam, influencing the vibrational and bending behaviors of the sensor (Supplementary Note 3). This centimeter-sized flexible sensor (Fig. 2c) can be directly integrated with the peripheral circuit through flexible printed circuit (FPC) connectors.

The sensor's performance was evaluated in various operation modes (Fig. 2d) by controlling the relative motion between the sensor and a test object. In the contact mode of static tactile sensation, vertical translation (velocity 1 cm s$^{-1}$) of the test object induced a downward deflection of the artificial antennae (displacement ~3 mm; tactile force ~30 mN). Following a short dwell time of 1 s, the object was retracted, resulting in the upward recovery of the artificial antennae. The sensor signal exhibited a sudden change followed by gradual recovery during contact and retraction (Fig. 2e). The profile of the sensor's piezoelectric response is similar to the adaptation behavior of "receptor potential" electrophysiologically recorded from insect tactile sensilla (Fig. 2a)[18,30]. The SA spike train (encoded by thresholding the input signal; Supplementary Note 4) contains long-duration (~1 s) pulses, while the FA spike train (encoded by detecting the changing rate of the input signal; Supplementary Note 4) includes short-burst (~0.1 s) pulses (Fig. 2e). Temporal profiles of the SA/FA spike trains briefly match the firing characteristics of biological SA/FA neurons. Reducing the deflection of the artificial antennae (*D*) during tactile contact results in a decreased response of the sensor signal (Fig. 2f). In the scanning mode of vibrotactile sensation, a test surface with periodic ridges (width 0.5 mm, height 1 mm) slid parallelly (velocity 2 mm s$^{-1}$) across the sensor. This operation induced repeated vibration of the artificial antennae, manifested by the short durations (170 ms, 90 ms) of SA/FA spike trains (Fig. 2g) and the high-frequency oscillations of sensor signal (Supplementary Note 1; Fig. S3).

Magneto-sensation of the sensor was also performed in the contact mode (dwell time of 1 s for contact) using a magnetic object instead (Fig. 2h). Due to the attractive magnetic interaction between the magnetic object and the sensor's magnetic appendage (Supplementary Note 5), a slight upward deflection of the artificial antennae, indicated by the gradual increase in the sensor signal, was observed (Fig. 2h). The two spike trains encoded from the sensor signal show that the magnetic-interaction-induced SA spike train precedes (0.56 s) the tactile-contact-induced FA spike train. Magnetic interaction between the sensor and the magnetic object allows magneto-sensation in the non-contact mode without tactile stimuli. During the approaching and retreating process of the magnetic source in a contactless manner, the sensor signal reveals that the artificial antennae underwent upward deflection followed by downward recovery (Fig. 2i), which differs from the case of contact-mode tactile sensation (Fig. 2f). Reducing the sensor-to-object distance (minimum value *L*) increased the sensor signal amplitude (Fig. 2i) since the magnetic interaction is distance-dependent. Multiple trails have been performed for each mode by measuring the contact force or the field strength, and the sensor shows high operational stability (cycling time >1000) and low detection limits (tactile force: 1.3 mN; surface pattern height: 50 μm; magnetic field: 9.4 mT) for mechano- and magneto-sensation (Fig. S4, Fig. S5). In contrast to existing bioinspired tactile sensors, our sensor displays distinct characteristics in both sensor structure and function, as well as signal encoding and processing (Supplementary Note 6).

## Artificial synaptic device

Sensory neurons play a crucial role in realizing essential information-processing functions in insects' antennal nerves. The ionic-gated synaptic transistor provides a hardware platform for emulating the synaptic functions of sensory neurons[31,32]. The artificial synaptic device was fabricated on a polyimide substrate through solution-processable fabrication of n-type semiconductors with a nanoflakes-on-film structure, deposition of electrodes with multiple planar gates, and coverage with an ion-gel layer as the gate dielectric (Fig. 3a, Fig. S6). The final device was obtained in flexible form as a pair (Fig. 3b). UV-Vis spectroscopy (Fig. S7), X-ray diffraction analysis (Fig. S7), and transmission electron microscopy (Fig. 3c, Fig. S7) confirm the existence of few-layer TMD nanoflakes and metal oxide. Characterization using atomic force microscopy (Fig. 3d), dark-field optical microscopy (Fig. S7), and scanning electron microscopy techniques (Fig. S8) reveal that the TMD nanoflakes (thickness 6 nm; size 50–100 nm) are distributed on the smooth surface of metal oxide film (thickness 40 nm). Surface potential mapping (Fig. S7) indicates that the nanoflakes adsorbed on the metal oxide film may facilitate charge trapping. Compared with the control device of a bare metal oxide film, the nanoflake-adsorbed device exhibits improvements in both transistor and synaptic characteristics (Fig. S10), which are beneficial for ionic gating and conductance modulation. By applying positive voltage spike trains to the planar gate, the cations (protons) in the ion-gel dielectric layer drifted and trapped on the interface between the ion gel and the semiconductor channel (Fig. S11). This resulted in the accumulation of carriers (electrons) in the channel surface and the formation of an electric double layer (Fig. S11) due to the electrostatic effect, as confirmed by the frequency-dependent, large specific capacitance (~1.5 μF cm$^{-2}$ at 1 kHz) of the ion gel (Fig. S12). Consequently, channel conductance increased, emulating the neurotransmitter release and excitatory facilitation of sensory neurons.

The synaptic behaviors of the device were then characterized. Spike-number dependent plasticity (SNDP) was examined by applying spike trains (50 Hz, 5 V) with different pulse numbers (Fig. 3e), revealing accumulative behavior in the device output. History-dependent plasticity of the device was investigated by applying four packets of spikes at various frequencies (25, 33.3, 50, 33.3 Hz). The device output showed potentiation with enhanced synaptic plasticity under high-activity states determined from spike history (Fig. 3f), aligning with the Bienenstock–Cooper–Munro (BCM) learning rule of biological neurons. The device exhibits high cycle-to-cycle stability in excitatory/inhibitory modulation under repeated stimuli of positive/negative spikes (Fig. 3g, Fig. S13). The operational stability of the device (bias voltage 10 mV) under repeated bending or after long-term storage (Fig. S13) further indicates its suitability for flexible electronics applications. Sensory memory of the insect nervous system is critical for cognitive and decision-making tasks. Here, we introduced sustaining spikes with small amplitude (ranging from 0 to 1.5 V) after the stimuli spikes to improve the retention behavior (increased from 8.0% to 73.9% at *t* = 20 s) of the device (Fig. 3h). This strategy enables the regulation of sensory memory in the device by adjusting the voltage of the sustaining spikes, offering adaptability for application-specific sensory processing with varying memory requirements. Furthermore, employing this strategy allows the modulation of memory retention behaviors across different devices, addressing concerns related to device-to-device variations (Supplementary Note 7, Fig. S14). The two planar gates of the device exhibited a similar gating effect (Fig. 3i), evidenced by the same peak current (1.2 μA). Simultaneously stimulating the dual gates resulted in a super-additive response (Fig. 3i) manifested as significantly enlarged output (7.1 μA), indicating that the device was sensitive to the temporal congruence of pairwise spikes (quantified by correlation coefficient; Supplementary Note 8). These device characteristics lay the foundation for recognizing the spatio-temporal patterns of multiple sensory inputs. In comparison with

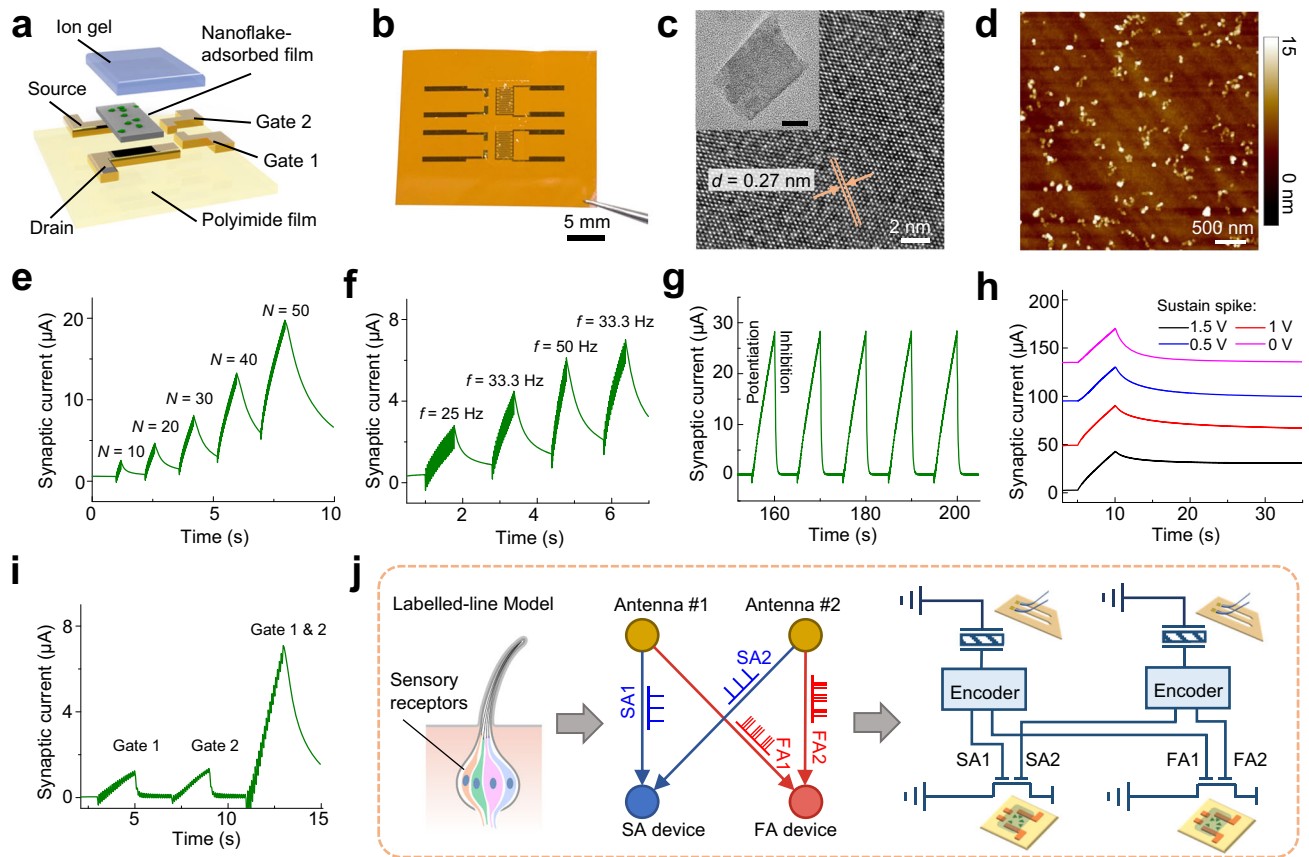

**Fig. 3 | Characterization and performance of the artificial synaptic device.**
**a**, **b** Exploded-view schematic illustration (**a**) and photograph (**b**) of the synaptic transistor with dual gates and nanoflake-adsorbed channel. **c** High-resolution TEM image of a nanoflake with high crystallinity. The measured lattice spacing (*d*) of 0.27 nm corresponds to the (100) plane of MoS$_2$. The TEM image in the inset (scale bar: 20 nm) shows the nanoflake's morphology. **d** AFM image of the semiconductor channel revealing that the nanoflakes are distributed on the smooth surface (root-mean-square roughness $R_q$ = 0.33 nm) of the metal oxide film. **e** Spike-number-dependent plasticity evaluated by applying spike trains with different spike numbers (*N* = 10, 20, 30, 40, 50). **f** History-dependent plasticity evaluated by applying spike trains with changing frequencies (*f* = 25, 33.3, 50, 33.3 Hz). **g** Cycle-to-cycle variation evaluated by repeatedly applying potentiation/inhibition spike trains (containing 50 positive and 50 negative pulses at 10 Hz). **h** Modulation of the sensory memory by applying sustaining spikes (reduced amplitude, same frequency) after stimuli spikes. **i** Gating effect of the dual gates. **j** Connection and signal flow of the sensor-device networks in the neuromorphic antennal sensory system imitate the "labeled-line" model of neuronal pathway and signal encoding in biological sensory neurons. Curves in (**h**) are offset vertically for clarity. SA1 and SA2: slowly adapting spikes from Antenna #1 and #2; FA1 and FA2: fast-adapting spikes from Antenna #1 and #2.

recently reported synaptic devices fabricated using solution-processable materials (Table S1), our device demonstrates advantages in synaptic and neuronal functions (demonstrated in the following part)[33–37].

In aggregate, the receptor cell population within the peripheral nervous system employs a "labeled-line" model of neuronal pathway and signal encoding (Fig. 3j). In this model, sensory neurons are specifically tailored to encapsulate and transmit different forms of sensory information along dedicated nerve fiber pathways[3,38,39]. Accordingly, we have integrated the sensor and device to build the neuromorphic antennal sensory system based on this biological model (Fig. 3j). Each of the two artificial antennae (receptor 1 and 2) generates an SA and an FA spike. The two pairwise SA spikes characterized by tonic firing behaviors are forwarded to the SA device (SA neuron), while the two pairwise FA spikes carrying phasic firing patterns are conveyed to the FA device (FA neuron). This configuration ensures that different sensory spikes are transmitted via separate pathways to distinct devices (Fig. 3j). Consequently, the static and dynamic attributes of spatiotemporal stimuli can be quantified through the synaptic currents recorded from the SA and FA device, respectively. Our system achieves device-level cognitive perception of sensory information in a multimodal, parallel, pulse-driven manner, distinguishing it from current artificial sensory systems that rely on external computational resources.

## Neuromorphic vibrotactile-perception

Capitalizing on the highly adaptive structures and sensorimotor functions of their antennae, ants demonstrate an acute ability to discern the texture and shape of their surroundings through antennal movement, akin to the whisking behaviors observed in rats (Fig. 4a). Remarkably, these perceptive abilities persist even in the absence of light, circumventing the need for sensory input from the visual system[2,28,40–42]. The temporal and spatial characteristics of tactile stimuli across various antennae are efficiently encoded and processed within the antennal nerve[30]. Similarly, our system implemented neuromorphic vibrotactile perception by parallelly scanning the electronic-antennae sensor across various test objects, including chess pieces, Braille codes, ridge patterns, and material textures (Fig. 4b). The encoded spikes of sensor signals were subsequently transmitted to the artificial synaptic devices for processing and recognition.

At first, chess profile classification was performed using a single artificial antenna aligned to the axial center of the chess piece. The acquired sensor signal reflects the lateral profile of the chess (maximum change ~3 mm), and the temporal patterns of the SA and FA spike trains reveal the deflection and vibration behaviors of the artificial antennae, respectively (Fig. 4c, Fig. S15). During the tactile contact process, the SA device's output displayed long cumulative features, while the FA device's output exhibited short bursting patterns (Fig. 4c,

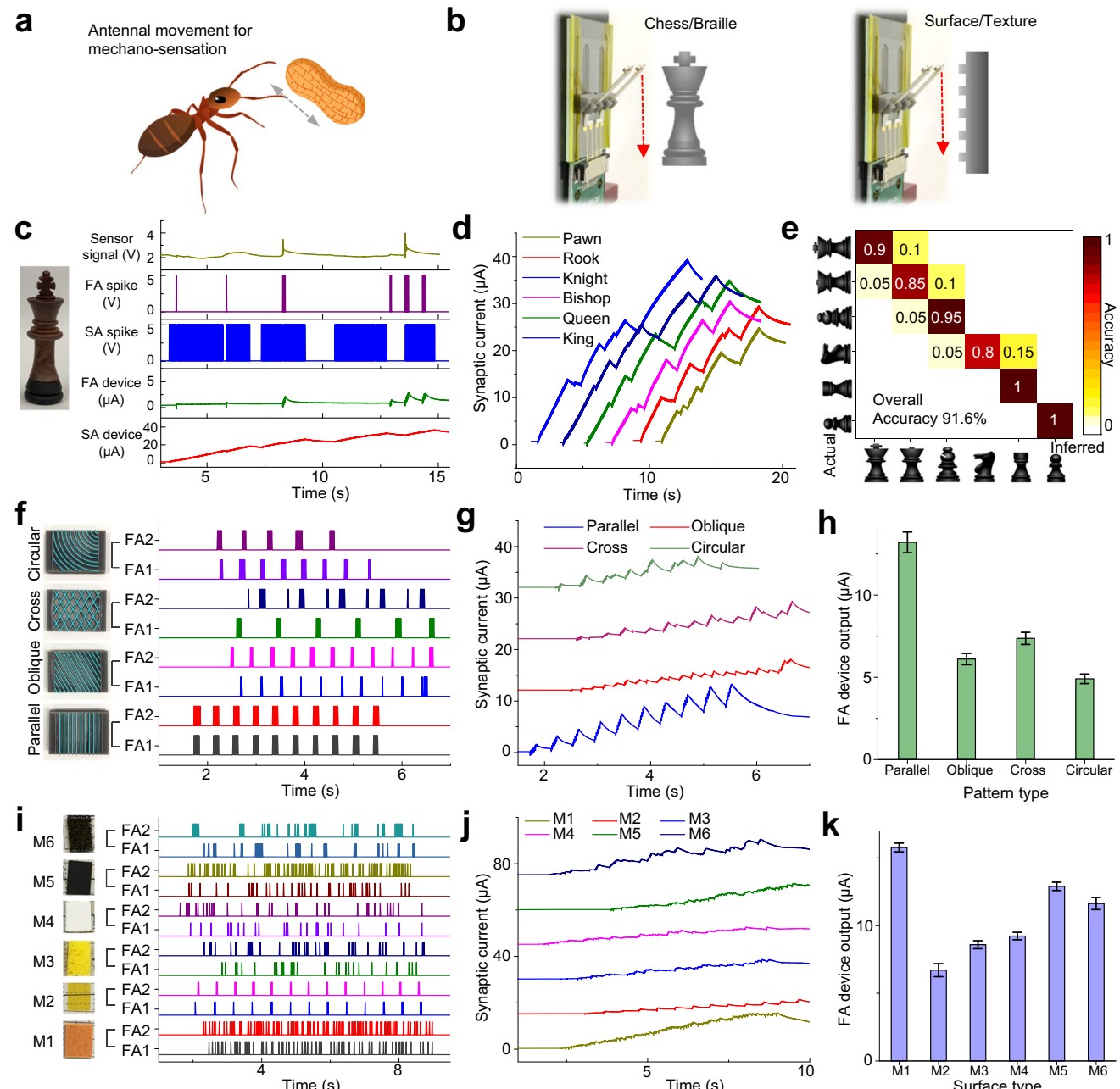

**Fig. 4 | Demonstration of neuromorphic vibrotactile perception. a** Illustration of antennal movement of the ant for mechano-sensation and tactile exploration. **b** Illustration of the experimental setup for the vibrotactile perception in profile classification and texture discrimination tasks. **c** Representative result of chess profile classification when a chess piece of "King" was scanned by a single artificial antenna. Time-resolved sensor signal, spike trains (SA and FA), and device output (SA and FA) were shown. **d** Time-resolved synaptic current of the SA device for all six types of chess pieces. **e** Results of chess profile classification shown as a confusion matrix. **f–h** Spatiotemporal patterns of the FA spike trains (**f**), time-resolved synaptic current of the FA device (**g**), and ending value of the FA device's output

measured at the end of a sensory event (**h**) when four different ridge patterns (guided by light blue lines) were scanned by a pair of artificial antennae. **i–k** Spatiotemporal patterns of the FA spike trains (**i**), time-resolved synaptic current of the FA device (**j**), and ending value of the FA device's output (**k**) when six different material textures (from M1 to M6) were scanned by a pair of artificial antennae. The error bars in (**h, k**) represent the standard deviation. The curves in (**d, g, j**) are offset vertically for clarity. FA1 and FA2: fast-adapting spikes from Antenna #1 and #2; M1–M6: metal foam, patterned plastic, dish sponge, canvas fabric, abrasive paper, and porous sponge.

Fig. S15). Such difference can be attributed to the chess's relatively smooth surface and the sensor's mainly tonic response. Given that the SA spikes with static characteristics better reflect the profile information, the SA device's synaptic currents for all six chess pieces were compared (Fig. 4d), revealing variations across all cases in the ending value of the SA device's output recorded at the end of a sensory event. Both the ending value of SA device's synaptic current and the mean firing rate of pairwise SA spikes (averaged throughout a sensory event)

can serve as classification criteria to identify different chess pieces. The classification results employing a decision-tree strategy (including training and inferring procedures; Supplementary Note 9) were presented as a confusion matrix (Fig. 4e), with elements on the diagonal signifying correct inferences. Notably, the overall recognition accuracy achieved using our system (91.6%) is close to the classification ability of human participants (average accuracy 87.3%; Table S2), who performed "blind" tactile exploration without visual cues under identical

conditions. Braille code identification was performed by parallelly sliding two artificial antennae across dot patterns (hemisphere, height 2 mm) created by 3D printing. Each artificial antenna was in contact with one column of Braille dots, and the pitch of dot columns (8 mm) was slightly smaller than that of the artificial antennae (10 mm), resulting in temporal delays in the sensor signals acquired from the two artificial antennae (Fig. S16). This temporal delay proves beneficial for recognizing symmetric spatial patterns. By evaluating the synaptic current of SA and FA devices (Fig. S17), the Braille codes corresponding to ten numbers (0-9) can be classified (Table S3).

Further experiments were performed to investigate the mechano-sensation performance in discerning delicate patterns and textures. Given that the FA spikes with dynamic characteristics better represent the vibration information, the output of FA device was utilized for recognition tasks. Initially, different ridge patterns (parallel, oblique, crisscross, circular) with a height of 1 mm were scanned by the two artificial antennae of the sensor (Fig. S18). The temporal profiles of pairwise FA spikes, encoded from two sensor signals, showed periodically bursting patterns corresponding to the spatial distribution of the ridges (Fig. 4f). Notably, the morphological differences between all four ridge patterns were reflected by temporal variations in the pairwise spike sequence (e.g., oblique pattern produced a 0.2 s lag between pairwise FA spikes). The output of the FA device (Fig. 4g) further revealed that the temporal congruence between the pairwise spikes significantly affected the synaptic current level. Specifically, the parallel ridge pattern, with the highest pairwise spike correlation coefficient (FA spikes) of 0.94, led to the largest synaptic current (FA device) of 13.2 μA (Table S4). Ridge patterns could be well classified based on the ending value of FA device's output acquired at the end of each sensory event (Fig. 4h). The influence of pitch or width of the ridges on surface pattern classification was evaluated experimentally (Fig. S19). Importantly, the movement speed of the sensor during surface scanning needs to be controlled within appropriate ranges (Supplementary Note 10, Fig. S20), since it affects the surface recognition accuracy. Moreover, vibrotactile perception of material texture (Fig. S21) was performed using various materials (metal foam, patterned plastic, dish sponge, canvas fabric, abrasive paper, and porous sponge). The results indicate that the pairwise FA spikes (Fig. 4i) briefly delineated texture information, and the material's roughness, porosity, and periodicity affected spatiotemporal patterns of the pairwise spikes. Time-resolved synaptic current of the FA device for all the materials (Fig. 4j) were compared. The ending value of FA device's output (Fig. 4k) and the mean firing rate of FA spikes (Fig. S22) exhibited discernable differences among all cases, enabling texture discrimination with high accuracy (overall classification accuracy of 93.3%; Fig. S23). Further enhancements in recognition accuracy can be achieved through optimization of tactile contact conditions and spike encoding strategies (Supplementary Note 11, Fig. S24).

### Neuromorphic magneto-perception

A hypothesized mechanism for insect magnetoreception suggests that ants potentially use their antennae to perceive the geomagnetic field, acting as a navigational compass for migration, orientation, and navigation (Fig. 5a)[19–27]. The directionality of this geomagnetic field exerts a significant influence on the magnetoreceptive behavior of ants. As proof of concept, the neuromorphic antennal sensory system was integrated into a mobile robot to showcase its capability of neuromorphic magnetoreception (Fig. 5b). Magnetic stimuli were essentially transformed into the gradual deflection of the artificial antennae due to the magnetically responsive characteristic of the sensor. The SA spikes characterized by slowly changing behaviors were utilized for magneto-perception. The mobile robot, equipped with the electronic-antenna sensor (installed upside down) on its front arm, sequentially executed a clockwise (360°) and a counterclockwise (−360°) rotational movement to detect the location of a target magnet (axially

magnetized) placed on the ground (Fig. 5b). During the "two-circle" rotation operation, the SA device generated two intense peaks corresponding to the minimum sensor-to-magnet distance when the robot aligned to the magnet twice. This experiment was repeated several times (sensor signals and encoded spikes shown in Fig. S25) by changing the magnet's location (from site 1 to site 6), which defines the desired heading direction (1/6 π, 3/6 π, 5/6 π, 7/6 π, 9/6 π, and 11/6 π) for the robot. The SA device's output (Fig. 5c) shows that the lag between the two peaks (changing from 9.5 s to 1.3 s), as well as the ending value of synaptic current (changing from 11.2 to 15.8 μA), varied among all cases (with the desired heading direction ranging from 1/6 π to 11/6 π), indicating angle-sensitive characteristics. Moreover, the mean firing rate of SA spike changed for each case, demonstrating angle-sensitive features. The polar plot (Fig. 5d) illustrates that the mean firing rate (SA spike) or the ending value of synaptic current (SA device) can clearly distinguish the desired heading angle, enabling robot navigation based on magnetic cues.

In our system, two magnetic appendages with opposite magnetization directions were mounted on the tips of two artificial antennae, forming an "anti-parallel" configuration that facilitates magneto-sensation of the magnetic properties of unknown materials. Material classification was performed by moving the sensor towards a test material (non-magnetic, magnetic, or ferrous) and subsequently retracting the sensor upon contact with the material (Fig. 5e). When in proximity to a magnetic material, the two artificial antennae deflected in opposite directions due to attractive and repulsive magnetic forces, respectively. The approaching of a ferrous material resulted in the upward defection of the two artificial antennae, both experiencing attractive magnetic interaction. In the case of non-magnetic material, magnetic interaction was absent while tactile contact force was applied, inducing downward deflection of the two artificial antennae. Various materials produced sensor signals with diverse profiles (Fig. 5e), and the encoded SA spikes derived from the sensor signal (Fig. 5f) elucidate the magnetic interaction process, resulting in distinct synaptic currents within the SA device (Fig. 5g). Consequently, material classification tasks can be successfully executed by analyzing the output of the SA device (Fig. 5h).

The magnetic interaction between the electronic-antennae sensor and other magnetic objects enables touchless interaction. This non-contact operation mode, designed to prevent cross-contamination and reduce infection, is suitable for smart interface applications. Herein, the system was built as an interactive device that enables non-contact interfacing with a human user wearing a magnetic-finger glove (Fig. 5i). The electronic-antennae sensor detects the magnetic stray field generated by the source magnet mounted on the glove, and the trajectory of finger motion affects the spatiotemporal patterns of the sensor signal. Different types of finger motions, including long click, double click, single click, up/down, and left/right, resulted in sensor signals (Fig. 5j) and encoded SA spikes (Fig. 5k) with distinct characteristics regarding temporal congruence, timing, and duration. Again, the SA spikes reveal the slowly changing sensory information of magnetic interaction, and the types of finger motion can be classified based on the ending value of the SA device's synaptic current (Fig. 5l, Fig. S26).

## Discussion

We present a neuromorphic antennal sensory system utilizing electronic antennae sensor and artificial synaptic device. By emulating the structural, functional, and neuronal features of ant antennae, our system achieves neuromorphic vibrotactile and magneto perception through multifunctional sensing, spike encoding, and synaptic processing. The system follows the labeled-line model of the receptor-neuron pathway, where SA and FA spike trains are separately sent to two synaptic transistors functioning as specialized neurons. This enables event-based, highly efficient, parallel processing of

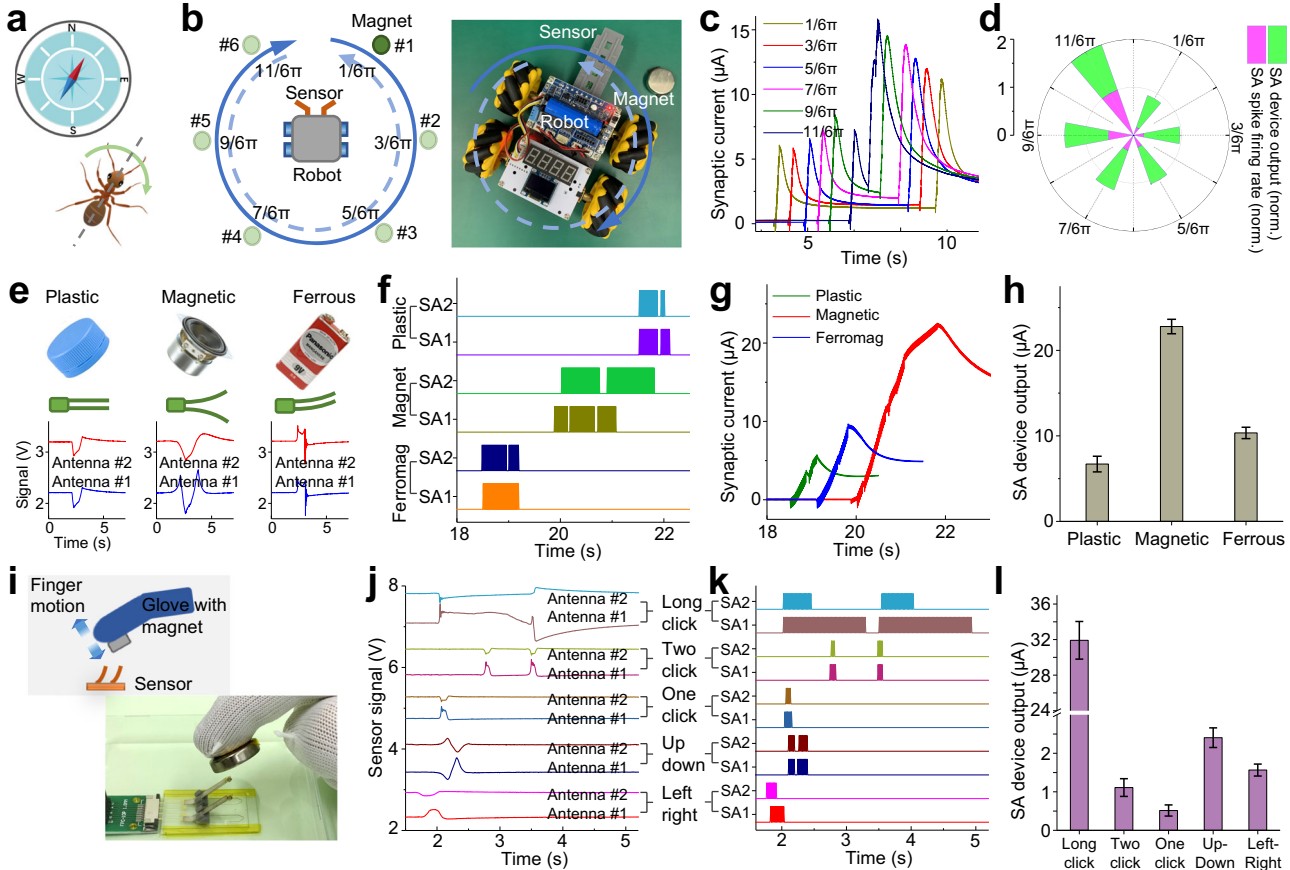

**Fig. 5 | Demonstration of neuromorphic magneto perception. a** Illustration shows that the ant uses its antennae during locomotion to perceive the geomagnetic field as a compass cue for navigation. **b** Experimental setup for the magnetic navigation of a mobile robot using artificial antennae pair. The robot executes rotational movement to detect the location of a target magnet, which is placed in different sites (from site 1 to site 6) corresponding to six desired heading directions (1/6 π, 3/6 π, 5/6 π, 7/6 π, 9/6 π, and 11/6 π). **c** Time-resolved synaptic current of the SA device corresponding to the six cases in (**b**). **d** Polar plot of the ending value of SA device's synaptic current and the mean firing rate of SA spikes, corresponding to the six cases in (**b**). **e** Time-resolved sensor signals of the artificial antenna pair for

classifying non-magnetic, magnetic, and ferromagnetic materials. **f**–**h** Spatiotemporal patterns of the SA spike trains (**f**), time-resolved synaptic current of the SA device (**g**), and ending value of the SA device's output (**h**) corresponding to the three cases in (**e**). **i** Schematic illustration and photograph of the touchless interfacing between a human user and the interactive device. **j**–**l** Time-resolved sensor signals of the artificial antenna pair (**j**), spatiotemporal patterns of the SA spike trains (**k**), and ending value of the SA device's output (**l**) for the classification of five different types of finger motions. The error bars in (**h**, **l**) represent the standard deviation. The curves in (**e**, **j**) are offset vertically for clarity. SA1 and SA2: slowly adapting spikes from Antenna #1 and #2.

spatiotemporal patterns of sensory stimuli. Our system can operate in different sensing modes that allow contact detection of profile and texture, as well as non-contact detection of magnetic/ferromagnetic material. Applications in profile classification, surface discrimination, material classification, magnetic navigation, and touchless interfacing were demonstrated to validate the cognitive intelligence in tactile and magneto sensations. Distinguishing itself from state-of-the-art artificial sensory systems (both tactile and visual), our system exhibits unique features in system architecture, sensor structure, signal processing, sensory functions, and neuronal characteristics (Table S5, Table S6, Supplementary Note 12)[6,8,12,13,15,43–45]. In our future work, we aim to integrate a flexible actuator with the sensor to enable antennal movement and active tactile exploration (Supplementary Note 13).

The insect antennae play fundamental roles in vibrotactile and magneto-perception, serving as a diminutive organ that facilitates active exploration of surfaces and objects through contact-based interactions, while enabling navigation and orientation in non-contact modes. These diverse sensory functions owe their efficiency to the antennal nerve, which is composed of receptor-neuron networks. These networks transmit spatiotemporal spikes of sensory stimuli, effectively deciphering and conveying environmental cues. By imitating the anatomical and neuronal characteristics of insect antennae, our

work addresses the challenging issues concerning tactile cognition, contactless perception, and sensory processing, and further expands the capability of mechano-perception to magneto-perception. The system developed here serves as an artificial sensory platform with neuromorphic, insectomorphic characteristics and hardware-level perceptual intelligence. It can be integrated with sensory robots and interactive devices, contributing to the development of augmented perception beyond human senses.

## Methods

### Fabrication of electronic-antennae sensor

The fabrication process began with laser cutting of a PET substrate (XF-162, XFnano Materials) to obtain a pair of antennae patterns (Fig. S27). The patterned substrate was cleaned with 2-propanol and deionized water, followed by the deposition of Ti/Au (5 nm/100 nm) through a shadow mask to form four electrodes (matched with an FPC connector). Piezoelectric films (thickness 52 μm, surface metalized PVDF, TE Connectivity), mechanically cut into 3 mm × 21 mm pieces, were aligned with the antennae patterns and laminated onto the topside of the substrate using double-sided adhesive tape (thickness 100 μm, Kapton). Each antennae pattern was deformed out-of-plane using a plastic ramp (45-degree slope), which was first fabricated by

stereolithography (SLA) 3D printing (Form3 printer, Formlabs) and then adhered onto the substrate surface using adhesive tape. This assembly procedure transformed the 2D patterns into 3D antennae structures. The plastic ramp, featuring a snap-fit design, securely fixed the bottom part of the artificial antennae while allowing the top part to deflect freely. Two tiny disc-shaped magnets (NdFeB, axial magnetization, diameter 2 mm, thickness 1 mm) were adhered to the tips of the artificial antennae (backside), and the magnetizations of the two magnets (referred to as "magnetic appendage") were maintained in opposite directions during installation. Both sides of the piezoelectric film were connected to the electrodes using conductive tape and conductive paste, yielding the final sensor as a pair.

## Fabrication of artificial synaptic device

Device fabrication began with the preparation of a metal oxide precursor by dissolving 0.4 g $SnCl_2$ (Macklin) in a 3 mL mixture of dimethylformamide (DMF) and ethanol (5:1, v/v) through stirring (300 RPM for one hour) and ultra-sonicating (10 min). The precursor solution was spin-coated (2000 RPM for 50 s) onto a polyimide substrate (thickness 150 μm, Mifang Electronics) that was cleaned by UV-ozone treatment (10 min). Note that the as-received polyimide was capped by two ultrathin protecting layers on both sides, facilitating easy handling of the flexible substrate during solution processing. Annealing in the air (300 °C for two hours) transformed the precursor material into a metal oxide film. Subsequently, this sample was treated with UV-ozone and then immersed in ethanol-water solution (45:55, v/v) containing semiconductor TMD nanoflakes (liquid exfoliated $MoS_2$ flakes dispersed in solution, concentration 18 mg $L^{-1}$, Graphene Supermarket) for one hour to enable the adsorption of TMD nanoflakes onto the surface of the metal oxide film. Then, the sample was rinsed with deionized water, blow-dried by $N_2$ gas, and annealed in an $N_2$ glovebox (100 °C for 30 min) to remove residues. Source, drain, and planar gate electrodes were deposited through a nickel shadow mask on the sample, defining an interdigitated channel (length 80 μm). Covering the regions of the channel and planar gates with an ion-gel layer prepared from biodegradable sodium alginate (Supplementary Note 14) completed the device fabrication. An additional transparent dressing (Tegaderm, 3 M) could be laminated to protect the device.

## System integration and signal encoding

The neuromorphic antennal sensory system was developed using an electronic-antennae sensor, two artificial synaptic devices, and a peripheral circuit (Fig. S28). Essentially, the system architecture mimics the "labeled-line" model of receptor-neuron pathways found in biological sensory systems. The piezoelectric signal from each sensor was initially converted into a voltage signal (similar to "receptor potential") through a charge-to-voltage converter. It was then encoded into SA spikes and FA spikes (Supplementary Note 15) using different coding strategies (pseudocode shown in Supplementary Note 4). The spike encoding process was fully implemented in a low-power microcontroller (ATmega328P) with a predefined sampling rate (200 Hz or 600 Hz, depending on task-specific resolution requirement). The two artificial antennae in the sensor produced two SA spikes and two FA spikes (typically 100 Hz). The pairwise SA and pairwise FA spikes, carrying the spatiotemporal information, were processed by two artificial synaptic devices (SA and FA devices), respectively. The synaptic currents generated by the two devices (SA and FA devices), along with the mean firing rates of the pairwise spikes (SA and FA spikes), were considered as the system outputs.

## Characterization and measurement

The semiconductor materials of the artificial synaptic device were characterized by optical microscopy (dark-field mode, DM500, Leica), scanning electron microscopy (MIFA-LMS, Tescan), atomic force microscopy (AFM and KPFM modes, Dimension Icon, Bruker), X-Ray

diffractometry (Ultima-IV, Rigaku), Raman spectroscopy (RMS1000, Edinburgh Instruments), and ultraviolet-visible spectroscopy (Cary-5000, Agilent). The ion gel of the artificial synaptic device was characterized using a semiconductor device parameter analyzer (B1500A, Agilent). Electrical measurements of the artificial synaptic device were performed using a semiconductor analyzer (4200A-SCS, Keithley). The devices can operate under a bias voltage as low as 10 mV. The performance of the electronic-antenna sensor was evaluated using a motorized force test stand (ESM301, Mark-10) equipped with a built-in force gauge.

## Mechanoreception and magnetoreception experiments

Mechanoreception experiments were performed by scanning the sensor's artificial antennae across the surface of a sample at constant velocity (typically 5 mm $s^{-1}$ for vibrotactile sensation). The sensor was vertically mounted to a translation stage that was installed on a motorized force test stand, and the sample was fixed to the moving probe of the force test stand, inducing relative motion between the sensor and the sample. During the scanning procedure, lateral sweep across the sample's surface caused deflection and vibration of the artificial antennae, and the synaptic current of the artificial synaptic device was recorded. A pre-deflection (typically 1 mm for vibrotactile sensation) was applied to the tip of the artificial antenna to ensure complete contact with the sample's surface. This scanning procedure was employed for the classification of chess pieces, Braille codes, surface patterns, and material textures.

The experiment setup for magnetoreception was customized for different tasks. In the robot navigation task, the sensor was mounted upside down to a cantilever arm installed on a mobile robot. The robot, equipped with four Mecanum wheels and an infrared receiver, can achieve stationary rotation under remote control. A target magnet (NdFeB, axial magnetization, diameter 3 cm, thickness 1 cm) was placed near the robot on the ground. The robot executed a clockwise rotation (360°) and then a counterclockwise (−360°) rotations to detect the location of the target magnet using the sensor. In the material classification task, the force test stand induced translational movement of a test material towards the sensor. The deflection of the artificial antennae was restricted to 1 mm during tactile contact with the test material. The test material included a plastic cap with a flat surface, an electric battery (9 V) with iron shells, and a cylindrical magnet (diameter 4 cm) with axial magnetization. In the finger-motion classification task, a human subject wearing a magnetic finger glove (with an axially magnetized thin magnet adhered to the fingertip's surface) performed designated finger motions over the sensor, and a plastic shield was placed above the sensor to prevent direct contact. Throughout all three tasks, the sensor's artificial antennae underwent deformation in a touchless manner due to the changing magnetic force applied to the sensor.

For classification tasks, the ending value of device outputs recorded at the end of a sensory event and the mean firing rate of pairwise spikes averaged throughout a sensory event were used as the classification criteria (Supplementary Note 9). Additionally, dynamic selection of the sensory spike suitable for sensory processing can be achieved by comparing the mean firing rates of the FA and SA spikes in the microcontroller.

## Human tactile perception experiment

The human tactile perception experiment was performed in a lightless environment, and the participant was allowed to perceive objects by using tactile cues without using visual cues. The objects used for the "blind" tactile test were six types of chess pieces (Fig. 4c), and the entire experiment involved procedures of training and recognition. During the training procedure, the participant was instructed to perceive a known chess piece (the corresponding type was disclosed) using one finger at the preferred speed and load (time limit 10 s). The

training procedure was repeated for each of the six types of chess pieces. Then, the recognition procedure started by asking the participant to identify the type of an unknown chess piece (time limit 10 s) through finger exploration. This procedure was repeated several times (≥ 50) using random chess pieces. Material texture classification followed the same procedures by replacing the chess pieces with texture samples (Fig. 4i). The adult human subjects who participated in the experiments were informed that they could disrupt the investigation at any time. The study has been conducted in accordance with the Declaration of Helsinki. Sex was not considered in this study, because no data relevant to sex has been collected.

Signed informed consent have been obtained from all the human subjects to perform the human tactile perception experiments. This work was approved by the Institutional Review Board at Nankai University with protocol no. NKUIRB2023098.

## Data availability
The data that support the findings of this study are available in the article, supplementary information file, source data file or from the corresponding authors upon request. Source data are provided with this paper.

## Code availability
The code for the spike-encoding function of the microcontroller is available at https://github.com/Jerix1989/E-antennae.git

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

## Acknowledgements

This study was supported by National Science Fund for Distinguished Young Scholars of China under Grant No. T2125005 (W.X.), National Natural Science Foundation of China under Grant No. 62201290 (C.J.), National Key R&D Program of China under Grant No. 2022YFE0198200, 2022YFA1204500, and 2022YFA1204504 (W.X.), Tianjin Science Foundation for Distinguished Young Scholars under Grant No. 19JCJQJC61000 (W.X.), and Shenzhen Science and Technology Project under Grant No. JCYJ20210324121002008 (W.X.). Special thanks to Prof. Shuang Hao (Northeastern University, China) for her invaluable assistance in revising the content related to biology and neuroscience. We express our gratitude to Junchi Liu, Qianbo Yu, Mingxin Sun, and Xu Ye (Nankai University, China) for their valuable contributions to the experiments. We also extend our thanks to Dr. Daniel Cheung for his insightful discussions.

## Author contributions

All authors participated in the discussions. W.X., K.T., and C.J. conceived the study. C.J. performed the fabrication, characterization, measurement, and analysis. H.X, L.Y., J.L., and Y.L. assisted in the fabrication. C.J. wrote the original draft. W.X. supervised the project. All authors reviewed and commented on the manuscript.

## Competing interests

The authors declare no competing interests.
