## [Peer Review File · Nature Communications]

REVIEWER COMMENTS

Reviewer #1 (Remarks to the Author):

This is an ambitious and interesting effort to build an entire biomimetic organ, from the sensor itself to the signal processing unit. I cannot comment on the technological aspects of it due to my own knowledge base, and restrict my own analysis to the bio-inspiration aspects, the biological understanding and the overall intro/discussion.

1. Overall, I do regret that the group has not teamed up with an insect neurobiologist/sensory physiologist who would master that part. Many aspects of the writing are weird if not wrong and it is too difficult for me to correct the entire text, nor is it the task of a referee. Specific examples include:

- line 44 nerves-> replace by nervous system

- all over: a neuromorphic electronic antennal nerve-> replace by neuromorphic antennal sensory system as you have also the sensor

- line 66 ".. vertical compression, lateral scanning, and magnetic proximity..." these represent different quantities but the ordering and the relationship between them is unclear.

2; Insect magneto-reception is a hot topic and there are very few works cited. The main papers (19,20) have been discussed at length. These authors do not claim to have found this sense in the antennae, it is a hypothesis.

3. The location of the sense in the antennae seems to be, if any, at the base of the antenna, near the Johnston's organ, and not at the tip as you have; So that is NOT bio-inspired

4. The antennae in the real animals tend to oscillate out of phase on purpose, you do not say anything about that because your antennae are passively reacting to the substrate; so that is NOT bio-inspired too. Overall, while I think there is enough "bioinspired" aspects in your work to warrant this claim, you need to discuss the limitation of your approach.

5. The comparison with existing bioinspired sensors is poor. Yes, it is obviously different from skin sensors, but how? what do we gain/lose? also, in terms of bioinspired sensory systems beyond the one you study (like for vision etc)- how does this compare?

Reviewer #2 (Remarks to the Author):

In the manuscript entitled "neuromorphic electronic antennae", the authors introduce the design of insect-inspired artificial sensory systems. Spatiotemporal sensory encoding and recognition are achieved, and vibrotactile and magneto perception capabilities are demonstrated. This work is interesting, and it

may inspire the field of bioinspired electronics and neuromorphic intelligence. There are several minor issues that need to be resolved before acceptance of the manuscript:

- (1) In figure 2f and figure 2i, how to differentiate between object tactile contact and non-contact magnetic proximity?
- (2) More explanation on the sensing mechanism of the sensor is needed. Is the sensor capable of identifying the bending direction?
- (3) Please elaborate on the significance and functionality of using synaptic transistors for sensory recognition, instead of using other methods such as machine-learning algorithm or Fourier signal analysis.
- (4) The knowledge or background of neuroscience should be explained to help the readers better understand the concepts. For instance, fast-adapting and slowly-adapting spikes in mechanoreceptors.
- (5) The authors are suggested to compare their works with other recent insect-inspired research works (Nature Nanotechnology, 2023, 18, 882-888).

Reviewer #3 (Remarks to the Author):

In this paper, the authors reported an electronic antennal nerve system comprising electronic antennae sensors and artificial synaptic devices inspired by an ants' antenna. The electronic antennae sensor, constructed from piezoelectric PVDF film, facilitates signal transmission through the integration of a coding circuit. Additionally, the encoded signals serve as input for integrated artificial synaptic devices based on ion gel. Based on this process, the electronic antennal nerve system demonstrates capabilities in vibrotactile and magneto perception. While the results are intriguing, certain aspects warrant further clarification.

1. There have been several reports on sensors using PVDF and synaptic devices utilizing ion gel (Journal of Materials Chemistry C 9, 8372-8394 (2021), ACS Applied Nano Materials 6, 1522-1540 (2023)). It would enhance the paper's informativeness to clarify the unique perspectives introduced beyond the combination of these devices into a neuromorphic electronic antenna.
2. In Figure S6, the authors deposited nanoflakes on the channel via the dip-coating method. However, atomic force microscopy images reveal random deposition, potentially leading to high device-to-device variations. Thus, the high device-to-device variation characteristics will likely cause high errors during classifications. The authors should investigate the device-to-device variation of the devices and its effect on classification accuracy.

3. In Figure 3j, the authors stated that both fast-adapting (FA) and slow-adapting (SA) are being utilized for the classification. However, it appears that only FA was used in Figure 4f and 4i, while only SA was used in Figure 5f and 5k. A clarification is needed regarding the specific reasons for choosing either FA or SA in these instances.

4. In Figure 4b, the authors analyzed the surface of the object by moving the device at a constant speed. In that case, the frequency or amplitude of the signal may vary with the movement speed of the device. Thus, the movement speed can affect the total accuracy of the classification, which should be further analyzed. It would be valuable if the authors to show the effect of movement speed on the output signals and their corresponding accuracy.

5. In Figure 4e, the recognition accuracy for the knight is lower compared to other chess pieces. Additionally, in Figure S21, recognition accuracy for M3 and M4 surfaces is lower than for other surfaces. The authors should provide an explanation for lower accuracy associated with specific objects or surfaces and propose potential avenues for improvement.

Response to Reviewers' Comments
(Manuscript ID: NCOMMS-23-49599-T)

Reviewer #1

Summary Comment from Reviewer #1:

This is an ambitious and interesting effort to build an entire biomimetic organ, from the sensor itself to the signal processing unit. I cannot comment on the technological aspects of it due to my own knowledge base, and restrict my own analysis to the bio-inspiration aspects, the biological understanding and the overall intro/discussion.

Response:

We express our sincere gratitude to the reviewer for providing thoughtful comments and constructive feedback aimed at improving the quality of our manuscript. As confirmed by the reviewer, our neuromorphic antennal sensory system tentatively emulates the sensory organ of insect antennae, in the aspects of sensor design, device functionality, and signal encoding and processing. In response to the reviewer's suggestion, we collaborated with a neurobiologist and revised the biology-related content under the guidance of this biologist. Additionally, we conducted an extensive literature survey to enhance the discussion on magnetoreception mechanisms and sensorimotor functions of the insect antennae. To address the reviewer's concern regarding active motion, we incorporated a discussion on the limitations associated with our sensor and proposed potential solutions. Moreover, we benchmark our neuromorphic antennal sensory system against state-of-the-art artificial sensory systems (particularly those related to visual perception) in the aspect of bio-inspired and insect-inspired perception. Lastly, we refined the language of the manuscript to enhance the clarity and readability. Throughout the revision process, our primary objective is to articulate the biological aspects of our work in a clear, understandable manner, emphasizing the design, the mechanism, the performance and of the bioinspired electronic system. Both the manuscript and accompanying supporting information have been extensively revised, with major modifications highlighted in RED. The point-to-point responses to the reviewer's comments are presented below.

Comment 1 from Reviewer #1:

Overall, I do regret that the group has not teamed up with an insect neurobiologist/sensory physiologist who would master that part. Many aspect of the writing are weird if not wrong and it is too difficult for me to correct the entire text, nor is it the task of a referee. specific examples include:

- line 44 nerves-> replace by nervous system
- all over: a neuromorphic electronic antennal nerve-> replace by neuromorphic antennal sensory system as you have also the sensor
- line 66 ".. vertical compression, lateral scanning, and magnetic proximity..." these represent different quantities but the ordering and the relationship between them is unclear.

Response:

We express our gratitude for the insightful comments provided by the reviewer.

In response to the reviewer's feedback, we have collaborated with a neurobiologist to enhance the biology-related content in the manuscript with the assistance of this scientist. Several concepts and terminologies pertaining to physiology and neuroscience have been modified to improve readability and ensure clarity for a diverse readership. Substantial revisions have been made to all sections pertaining to biology and neuroscience. We have also improved the content related to the flexible sensor and synaptic device in the manuscript to ensure that the structural and functional characteristics as well as the mechanistic interpretations of our neuromorphic antennal sensory system are presented in a clear and understandable manner, as they are the most important part in our work.

Below are the specific modifications related to biology and neuroscience:

- Minor Modification #1:
 - Original (Title): Insect-inspired neuromorphic electronic antennae
 - Revised (Title): **Neuromorphic antennal sensory system**
- Minor Modification #2:
 - Original (Introduction): "previous efforts to emulate natural tactile sensory organs and nerves"
 - Revised (Introduction): "previous efforts to emulate natural tactile sensory organs and **nervous systems**"
- Minor Modification #3:
 - Original (throughout the manuscript): "neuromorphic electronic antennal nerve"; "electronic antennal nerve system"; "insect-inspired electronic antennal nerve"
 - Revised (throughout the manuscript): "**neuromorphic antennal sensory system**"
- Minor Modification #4:
 - Original (Introduction): "The electronic-antennae sensor in this system is designed with biomimetic, flexible, three-dimensional (3D) structures, and the sensor's responses to vertical compression, lateral scanning, and magnetic proximity are studied."
 - Revised (Introduction): "**The electronic-antennae sensor in this system features biomimetic, flexible, three-dimensional (3D) structures. The sensor's responses to vertical compression, lateral scanning, and magnetic proximity are separately measured in different operation modes to assess its sensing capabilities for pressure, vibration, and magnetic stimuli.**"

- Minor Modification #5:
 - Original: “employs the encoding strategy of fast adapting (FA) and slowly adapting (SA) mechanoreceptors”
 - Revised: “employs the encoding strategy of fast-adapting (FA; sensitive to dynamic stimulation) and slowly-adapting (SA; sensitive to static stimulation) mechanoreceptors”

- Minor Modification #6:
 - Original (Sub-title): “Electronic antennal nerve inspired by insects”
 - Revised (Sub-title): “Neuromorphic antennal sensory system”

- Minor Modification #7:
 - Original (main content): “the peripheral nerve”
 - Revised (main content): “the peripheral nervous system”

- Major Modification #1:

Abstract: “The antennae in insects facilitate the nuanced detection of vibrations, deflections, and the non-contact perception of magnetic or chemical stimuli, capabilities not found in mammalian skin. The advent of insect-inspired artificial sensory systems may pave the way for perceptual augmentation that transcends human sensory capabilities, presenting unexplored possibilities in sensing technology. Here, we report a neuromorphic antennal sensory system (electronic antennae) that emulates the structural, functional, and neuronal characteristics of ant antennae.”

- Major Modification #2:

Introduction, paragraph 1: “Biological tactile sensory organs, encompassing skin, whiskers, antennae, among others, have manifested in diverse forms, each possessing unique anatomical structures, sensory functions, and neuronal encoding or processing mechanisms. Mainly, these organs demonstrate proficiency in mechano-sensation related to pressure and vibration, utilizing spatiotemporal encoding methods to interpret somatosensory information obtained from various mechanoreceptors. These intricate systems endow a refined perception of textures, profiles, and shapes during both tactile interaction and active exploration.”

- Major Modification #3:

Introduction, paragraph 2: “Insects' tactile sensory organs, despite their diminutive scale and paucity of neurons compared to mammals, exhibit efficient processing and multimodal sensory functions encompassing mechano-perception, magneto-perception, audio-perception, and chemo-perception.^{2,16,17} These capabilities could serve as blueprints for the development of biomimetic sensory platforms. Hallmarked by their segmented, flexible, three-dimensional architecture, insect antennae deliver exceptional mechano-sensory performance in response to deflections and vibrations.¹⁸ Some research grounded in cellular and molecular evidence tentatively hypothesizes that specific insects, such as ants, enable magneto-reception via their antennae's mechanically sensitive, magnetite-infused magnetoreceptors.¹⁹⁻²⁷ The exquisite antennal structures, densely innervated with sensory receptors and neurons, emit spatiotemporally-encoded neural spike sequences, allowing for the detection of vibrotactile and magnetic stimuli. The perceptual acuity is comparable to, or even surpasses, that of human skin, thereby enabling insects to execute complex tasks, including foraging, object identification, and navigation.^{2,28,22,29}”

- Major Modification #4:

Main content: “In a vast array of insects, encompassing flies, bees, and ants, antennae emerge as intricately complex, multifunctional sensory apparatuses. Antennae underpin various sensory modalities, including mechanoreception, magnetoreception, audio-perception, and chemoreception.^{2,17,22,30} Examining the case of ants, these creatures exploit their paired antennae to achieve tactile and magneto-perception with notable acuity (Fig. 1a). The segmented structure of these antennae functions as a sensorimotor organ that can actively maneuver for mechanosensation of contact, vibration, and surface texture.¹⁶ Furthermore, certain literature hypothesizes that the ion-rich particles present within the antennae potentially facilitate the ant's perception of the earth's magnetic field via the detection of the magnetic force applied on the antennae.¹⁹⁻²⁷ The antennae are replete with diverse sensory receptors and neurons, comprising what is known as Johnston's organ. These components serve to identify and transcode movements of the antennae into slowly-adapting (tonic firing for sustained stimuli) and fast-adapting (phasic firing for changing stimuli) neural spikes (Fig. 1b).¹⁸ This orchestration of spatiotemporal spikes, carrying sensory information, is subsequently processed in the central nervous system, facilitating multifunctional perception.^{3”}

- Major Modification #5:

Main content: “A close examination of the ant's antenna divulges a multi-segment, elbowed structure that optimizes mechano-sensation (Fig. 2a). The binding of the short basal segment of the scape, characterized by its limited movement, and the elongated distal segment of funiculus, known for its unrestricted movement, embodies a pliant joint structure. This structure, innervated to the mechano-responsive sensory receptors, empowers both active tactile exploration and passive tactile sensation.^{2,18”}

- Major Modification #6:

Main content: “In aggregate, the receptor cell population within the peripheral nervous system employs a “labeled line” model of neuronal pathway and signal encoding (Fig. 3j). In this model, sensory neurons are specifically tailored to encapsulate and transmit different forms of sensory information along dedicated nerve fiber pathways.^{3,38,39”}

- Major Modification #7:

Main content: “Capitalizing on the highly adaptive structures and sensorimotor functions of their antennae, ants demonstrate an acute ability to discern the texture and shape of their surroundings through antennal movement, akin to the whisking behaviors observed in rats (Fig. 4a). Remarkably, these perceptive abilities persist even in the absence of light, circumventing the need for sensory input from the visual system. The temporal and spatial characteristics of tactile stimuli across various antennae are efficiently encoded and processed within the antennal nerve.^{2,28,40,41,42”}

- Major Modification #8:

Main content: “A hypothesized mechanism for insect magnetoreception suggests that ants potentially use their antennae to perceive the geomagnetic field, acting as a navigational compass for migration, orientation, and navigation (Fig. 5a).¹⁹⁻²⁷ The directionality of this geomagnetic field exerts a significant influence on the magnetoreceptive behavior of ants.”

- Major Modification #9:

Conclusion: “The insect antennae play fundamental roles in vibrotactile and magneto-perception, serving as a diminutive organ that facilitates active exploration of surfaces and objects through contact-based interactions, while enabling navigation and orientation in non-contact modes. These diverse sensory functions owe their efficiency to the antennal nerve, which is composed of receptor-neuron networks. These networks transmit spatiotemporal spikes of sensory stimuli, effectively deciphering and conveying environmental cues.”

Comment 2 from Reviewer #1:

Insect magneto-reception is a hot topic and there are very few works cited. The main papers (19,20) have been discussed at length. These authors do not claim to have found this sense in the antennae, it is a hypothesis.

Response:

We express our appreciation to the reviewer for their valuable insights into insect magnetoreception.

First, in response to the reviewer's suggestion, we conducted a comprehensive literature survey on animal magnetoreception, particularly focusing on the origin and mechanism involved. In total, we examined nine reference articles listed below, of which the first five have been added to the reference list of the revised manuscript, while the last four were already include in the original manuscript. Notably, the first reference article shown below was featured on the cover of *Nature* in June 2021, underscoring the significance and impact of the magnetoreception research, as highlighted by the reviewer.

- Magnetic sensitivity of cryptochrome 4 from a migratory songbird. *Nature* **594**, 535-540 (2021).
- Unravelling the enigma of bird magnetoreception. *Nature* **594**, 497-498 (2021).
- Long-distance navigation and magnetoreception in migratory animals. *Nature* **558**, 50-59 (2018).
- A magnetic protein biocompass. *Nature Materials* **15**, 217-226 (2016).
- A putative mechanism for magnetoreception by electromagnetic induction in the pigeon inner ear. *Current Biology* **29**, 4052-4059 (2019).
- Ant antennae: are they sites for magnetoreception? *Journal of The Royal Society Interface* **7**, 143-152 (2010).
- Magnetic anisotropy and organization of nanoparticles in heads and antennae of neotropical leaf-cutter ants, *Atta colombica*. *Journal of Physics D: Applied Physics* **47**, 435401 (2014).
- Identifying cellular and molecular mechanisms for magnetosensation. *Annual review of neuroscience* **40**, 231-250 (2017).
- Magnetoreception in Hymenoptera: importance for navigation. *Animal Cognition* **23**, 1051-1061 (2020).

The surveyed literature reveals the elusive nature of the mechanisms employed by animals to sense magnetic field. Presently, three leading hypotheses for magnetoreception include electromagnetic induction (e.g. in pigeons), magnetite-based magnetoreception (e.g., in bacteria and insects), and biochemical magnetoreception (e.g., in migration birds). It is crucial to note that all these mechanisms are speculative,

because identifying magnetoreceptors is inherently challenging and magnetic fields freely permeating biological tissues are hard to manipulate. Concerning insects, the origin and sensory organ of magnetoreception remain enigmatic, although its confirmed importance in migration, orientation, and navigation is well established. In the case of migratory ants, a putative hypothesis suggests that the substantial amount of magnetic material associated with Johnston's organ and other joints in the antennae potentially produce a mechanosensory output modulated by the magnetic field, forming the basis of the magnetic sense. Note that Johnston's organ is a collection of sensory cells located in the pedicel (second segment) of the antennae in insects. However, the internal transduction mechanisms and the comprehensive utilization of magnetic information by insects lack clear understanding and compelling evidence. Knowledge and information regarding the location and composition of magnetic compasses and associated neural mechanisms, encompassing sensing, transduction, storage, and retrieval of magnetic stimuli, remains incomplete.

In response to these insights, we have revised the manuscript's content pertaining to the mechanism and location of magnetoreception. It is emphasized in the manuscript that the proposed explanations for magnetoreception, based on existing literature, remain hypothetical.

The reviewer may refer to the following revised content in the manuscript. Note that one sentence has been deleted for clarity:

- Modification #1: “Some research grounded in cellular and molecular evidence tentatively hypothesizes that specific insects, such as ants, enable magneto-reception via their antennae's mechanically sensitive, magnetite-infused magnetoreceptors.”
- Modification #2: “Furthermore, certain literature hypothesizes that the ion-rich particles present within the antennae potentially facilitate the ant's perception of the earth's magnetic field via the detection of the magnetic force applied on the antennae.”
- Modification #3: (deleted) “~~Fine-grained magnetic crystals in the antennae exhibiting high magnetic anisotropy enable magnetic field modulated mechano-sensation, which underlies the magneto-perception in ants.~~”
- Modification #4: “A hypothesized mechanism for insect magnetoreception suggests that ants potentially use their antennae to perceive the geomagnetic field, acting as a navigational compass for migration, orientation, and navigation (Fig. 5a). The directionality of this geomagnetic field exerts a significant influence on the magnetoreceptive behavior of ants.”

The reviewer may also refer to the added references in the manuscript:

- Reference 23-27.

Comment 3 from Reviewer #1:

The location of the sense in the antennae seems to be, if any, at the base of the antenna, near the Johnston's organ, and not at the tip as you have; So that is NOT bio-inspired

Response:

We express our gratitude to the reviewer for providing insightful comments on the magnetic sense.

First of all, to address the reviewer's suggestion, we examined the location of magnetic sense by referring to the relevant articles in our reference list. For ant antennae, the characterization results from the literature articles elucidate that ion-rich materials (specifically magnetite crystals in particle form) are situated in the antennal joints and Johnston's organ, close to the basal part of the antenna, rather than at the distal end. A putative mechanism of magnetoreception in antennae hypothesized that the magnetic moments of these magnetic crystals under an external magnetic field can generate a mechanical moment detected by mechano-sensitive structures such as Johnston's organ (serving as a multimodal mechanosensitive sensory organ, Johnston's organ also facilitates the detection of antennal vibrations or deflections induced by gravity, wind or touch).

Regarding our electronic antennae sensor, despite that the sensor's magnetic appendage is positioned at the tip of the artificial antennae, the operational principle of the sensor is similar to the hypothesized mechanism of magnetoreception. Our sensor essentially converts the applied magnetic field into mechanical deformation of a mechano-sensitive piezoelectric film with flexible cantilever structure. In principle, our sensor is bioinspired by the emulation of antennae magnetoreception, although mechanistic explanation of insect magnetoreception remains speculative. Furthermore, in comparison with state-of-the-art magnetic e-skin sensors employing spintronic principles (magnetic moment of electron spin affected by applied magnetic field) and demonstrating remarkable sensitivity, our electronic antennae sensor emulates the magnetoreception principle while offering advantages such as spatiotemporal recognition, multimodal integration, and cost-effective scalable fabrication.

In response to the reviewer's suggestions, we have carefully revised the content related to the electronic antennae sensor, notably reducing the use of terms like "bioinspired" and "biomimetic". Our aim is to ensure that the explanations of the magnetoreception mechanism in antennae and magnetic perception principle of our sensor are accurate.

The reviewer may refer to the following revised content in the manuscript:

- Modification #1:
 - Original (Title): Insect-inspired neuromorphic electronic antennae
 - Revised (Title): **Neuromorphic antennal sensory system**

- Modification #2:
 - Original (Sub-title): "Electronic antennal nerve inspired by insects"
 - Revised (Sub-title): "**Neuromorphic antennal sensory system**"

- Modification #3:
 - Original (Sub-title): "Biomimetic electronic-antennae sensor"
 - Revised (Sub-title): "**Electronic-antennae sensor**"

- Modification #4:

- Original (Introduction): “In this work, we report an insect-inspired electronic antennal nerve for vibrotactile- and magneto-perception.”
- Revised (Introduction): “In this work, we report a neuromorphic antennal sensory system designed for vibrotactile- and magneto-perception.”
- Modification #5:
 - Original (main content) “Besides, the ant antennae contain a high concentration of ion-rich particles, allowing the ant to perceive the earth’s magnetic field by sensing the magnetic force exerted on the antennae. The antennae are innervated by multiple sensory receptors and neurons (forming Johnston’s organ), which detect and encode the antennal movements into slowly adapting (tonic firing) and fast adapting (phasic firing) neural spikes (Fig. 1b).”
 - Revised (main content): “Furthermore, certain literature hypothesizes that the ion-rich particles present within the antennae potentially facilitate the ant's perception of the earth's magnetic field via the detection of the magnetic force applied on the antennae. The antennae are replete with diverse sensory receptors and neurons, comprising what is known as Johnston's organ. These components serve to identify and transcode movements of the antennae into slowly-adapting (tonic firing for sustained stimuli) and fast-adapting (phasic firing for changing stimuli) neural spikes (Fig. 1b)”
- Modification #6:
 - Original (main content): “Inspired by the ant antennae’s structural, functional, and neuronal characteristics, an electronic antennal nerve system was developed by integrating an electronic antennae sensor, a spike-encoding circuit, and artificial synaptic devices.”
 - Revised (main content): “By emulating the structural, functional, and neuronal characteristics of ant antennae, a neuromorphic antennal sensory system was developed using an electronic antennae sensor, a spike-encoding circuit, and artificial synaptic devices.”
- Modification #7:
 - Original (caption of Figure 1): “Design of the insect-inspired electronic antennal nerve”
 - Revised (caption of Figure 1): “Design of the neuromorphic antennal sensory system”
- Modification #8:
 - Original (conclusion): “Inspired by ant antennae’s structural, functional, and neuronal features, the system can achieve neuromorphic vibrotactile and magneto perception through multifunctional sensing, spike encoding, and synaptic processing.”
 - Revised (conclusion): “By emulating the structural, functional, and neuronal features of ant antennae, our system achieves neuromorphic vibrotactile and magneto perception through multifunctional sensing, spike encoding, and synaptic processing.”

Comment 4 from Reviewer #1:

The antennae in the real animals tend to oscillate out of phase on purpose, you do not say anything about that because your antennae are passively reacting to the substrate; so that is NOT bio-inspired too. Overall, while I think there is enough "bioinspired" aspects in your work to warrant this claim, you need to discuss the limitation of your approach.

Response:

We express our gratitude to the reviewer for this insightful comment on the active movement and sensorimotor function of antennae.

In response to the reviewer's comment, we conducted a literature survey on the topic of antennal movement, antennal sensorimotor function, and active tactile exploration. The examined reference articles listed below have significantly contributed to our understanding of these aspects.

- Sensorimotor ecology of the insect antenna: active sampling by a multimodal sensory organ. *Advances in Insect Physiology* **63**, 1-105 (2022).
- Mechanosensation and adaptive motor control in insects. *Current Biology* **26**, R1022-R1038 (2016).
- Antennal movements and mechanoreception: neurobiology of active tactile sensors. *Advances in Insect Physiology* **32**, 49-205 (2005).
- Active touch in orthopteroid insects: behaviours, multisensory substrates and evolution. *Philosophical Transactions of the Royal Society B: Biological Sciences* **366**, 3006-3015 (2011).
- Antennal mechanosensors and their evolutionary antecedents. *Advances in Insect Physiology* **49**, 59-99 (2015).
- Both stiff and compliant: morphological and biomechanical adaptations of stick insect antennae for tactile exploration. *Journal of the Royal Society Interface* **15**, 20180246 (2018).

These research reports underscore the active and purposeful movement of insect antennae, as mentioned by the reviewer (the antennae can move on purpose). Antennae share many morphological, neurobiological, and functional traits with the leg, since mechanosensory feedback is utilized in guiding the movement in both of them. The active movement capabilities and multimodal sensory functions of insect antennae play a pivotal role in various behaviors, including spatial orientation, search, tactile exploration, and communication. Depending on whether the size and structure of the antennae allow active sampling of external surfaces or objects through physical contacts, antennae can be categorized into contact antennae (e.g., in stick insects, ants, and honeybees) and non-contact antennae (e.g., in fly, mosquito, moth).

However, it is pertinent to note that our work primarily focuses on the perception functions of the neuromorphic antennal sensory system rather than motion control functions. The sensing experiments in our work are performed by passive movement to independently evaluate the tactile / magnetic sensation capabilities without using active exploration or active motion. This approach is justified by our system's intended applications in mobile robots or the human bodies, which inherently possess self-motion capabilities. Nevertheless, we recognize potential technical limitations in our current system regarding sensorimotor functions and active tactile exploration performance, and these limitations include: lack of adaptive control of the posture of the artificial antennae, unable to finely change the contact depth during tactile contact, and unable to achieve active touch or active contact. To address these limitations and develop the active motion capabilities, we propose a feasible strategy involving three key improvements to our current system:

(1) Geometry Optimization: The geometry of the electronic antennae sensor needs to be reduced and redesigned to meet the requirements of active tactile exploration tasks. Adjustments to aspect ratio and total length are essential.

(2) Integration of Flexible Actuator: A flexible actuator needs to be well integrated into the artificial antennae to facilitate active movement, such as bending. Soft or flexible materials with stimuli-responsive behavior, such as thermo actuating, electric actuating, or piezoelectric actuating, can be explored for this purpose.

(3) Closed-Loop Motion Control: Develop a closed-loop motion control method for the sensor array by drawing inspiration from the adaptive motor control strategies observed in insect antennae. This control method aims to ensure the reliable and efficient acquisition of mechanosensory information, which is further used to guide the active movement of the artificial antennae.

As suggested by the reviewer, we have addressed the limitation of the current approach regarding active motion in the revised supplementary information. Additionally, in the revised manuscript, we have outlined our vision for future work, specifically focusing on emulating the sensorimotor functions of insect antennae.

The reviewer may refer to the following revised content in the supplementary information:

- **Supplementary Note 13. Limitations on active motion.**

The reviewer may also refer to the following revised content in the manuscript:

- **“In our future work, we aim to integrate a flexible actuator with the sensor to enable antennal movement and active tactile exploration (Supplementary Note 13).”**

Comment 5 from Reviewer #1:

The comparison with existing bioinspired sensors is poor. Yes, it is obviously different from skin sensors, but how? what do we gain/lose ? also, in terms of bioinspired sensory systems beyond the one you study (like for vision etc)- how does this compare?

Response:

We would like to express our gratitude to the reviewers for this helpful comment.

In response to the reviewer's suggestions, we compare our sensor with existing bioinspired flexible sensors. Subsequently, we elucidated the distinctive features and advantages of our sensor, specifically focusing on sensor structure, sensing functions, and sensory processing. Finally, we benchmark our neuromorphic antennal sensory system against the state-of-the-art artificial sensory systems (primarily visual systems) in the aspect of bio-inspired and insect-inspired perception.

Comparison with Existing Bioinspired Flexible Sensors

We systematically reviewed recent articles on bioinspired flexible sensors, focusing on tactile and magnetic sensation (Supplementary References S15-S25). These sensors predominantly draw inspiration

from the tactile sensory organ of mammalian skin, often featuring a multilayered planar or out-of-plane hairy structure. While some flexible sensors in these works employ a three-dimensional structure to enable adaptive sensation, their fabrication involves high-cost, complicated manufacturing processes such as photolithography and micropatterning. Moreover, despite the utilization of functional materials with excellent sensing capabilities, such as carbon nanotubes, nanofibers, triboelectric material, and piezoelectric materials, the tactile sensing functions are generally limited to pressure, temperature, friction, and bending detection. Vibrational and non-contact tactile sensation (such as magnetically responsive tactile perception) are rarely explored in these bioinspired sensors. Regarding sensory processing and sensory encoding, the state-of-the-art flexible sensors recently reported in the literature (e-skin, magnetosensitive e-skins, biomimetic whisker) can achieve high sensitivity and fine spatial resolution in detecting sensory stimuli. However, in most cases, the acquired sensing signals are processed in digital form using external computational and memory resources (for instance, computer). Although some works employ the spike-coding method for signal processing, sensory processing is typically implemented through machine-learning algorithms or the Fourier transform method. Therefore, these reported flexible sensors, along with their sensory systems, have made limited attempts to implement bioinspired signal encoding and processing using biologically plausible methods (e.g., synaptic processing) or brain-inspired hardware (e.g., synaptic transistors).

In comparison with currently reported bioinspired flexible sensors, our electronic antennae sensor boasts unique characteristics and advantages as outlined below.

(1) Sensor Structure and Function: our electronic antennae sensor features segmented, flexible, three-dimensional structures and multifunctional sensing capabilities, including vibrotactile and magneto perception. Emulating the antennal sensory system of insects, our sensor breaks the design constraints of current electronic skin sensors inspired by the sensory organ of skin. In addition, our sensor offers various operational modes for detecting stimuli including deflection, vibration, and magnetic field with low detection limit, thus enabling advanced applications such as profile detection, texture discrimination, Braille code recognition, material classification, magnetic navigation, and touchless interfacing. These versatile functions and applications are hard to achieve using other bioinspired tactile sensors.

(2) Signal Encoding and Processing: our sensory system employs spike-based signal encoding strategies (fast-adapting and slowly-adapting coding) inspired by mechanoreceptors. Furthermore, our sensory system comprising electronic antennae sensors and synaptic devices creates a 2-by-2 receptor-neuron network that emulates the “labeled-line” model of the biological receptor-neuron pathway. This design facilitates the separate processing of static and dynamic signals. Achieving device-level cognitive perception of spatiotemporal sensory information in a parallel, energy-efficient manner, our system excels in synaptic processing with neuromorphic intelligence. Additionally, important neuronal functions, including spatiotemporal integration, multimodal sensory processing, and controllable sensory memory, are realized in the synaptic device. These advancements contribute to the achievement of vibrotactile and magneto perception, which are scarcely attempted in other artificial sensory systems. Notably, spike encoding, synaptic processing, and neuronal functions are implemented in the neuromorphic hardware and the microcontroller unit, making our system suitable for wearable applications compared to other bioinspired sensory systems

relying on external computing or memory resources (such as CPU, GPU, or DSP processors).

Comparison with Bio-Inspired Visual Sensory Systems

Finally, we compare our work with recent advancements in insect-inspired and bio-inspired artificial sensory systems (Supplementary Reference S26-S32). Typically, the majority of cutting-edge artificial sensory systems are designed as visual sensory systems, drawing inspiration from mammalian eyes or insect compound eyes, wherein the sensory input is limited to visual signals. These systems often utilize planar structures on rigid silicon or spherical structures on flexible substrates, and the lack of three-dimensional structures with flexible or adaptive designs poses a limitation. Moreover, the fabrication of these systems involves chemical or physical deposition and other complicated manufacturing procedures to grow functional materials, so the fabrication cost is high. Emerging artificial visual sensory systems, utilizing in-sensor or near-sensor computing architecture, can incorporate both sensing and processing functions and simplify the system layout and fabrication. However, these systems still depend on silicon fabrication and packaging methods to create integrated sensory-computing architectures with interconnections. In terms of sensing functionalities, these visual systems can achieve various functions of visual perception, such as motion and movement perception (inspired by the fly's compound eye), panoramic and amphibious imaging (inspired by the crab's compound eye), wide field-of-view detection (inspired by the locust's compound eye), and scotopic/photopic adaptation (inspired by the retina). Certain bioinspired visual sensory systems implement visual perception functionalities at the device level using memristors or phototransistor arrays to enable sensory processing or sensory memory. Their potential applications are predominantly related to machine vision or computer vision, encompassing tasks such as motion detection, movement recognition, image recognition, and color perception.

In contrast, our neuromorphic antennal sensory system has several key features that distinguish it from the recently reported artificial visual sensory systems.

- (1) **Multimodal sensory input:** our system is inspired by the sensory organ of the insect's antennae, utilizing multimodal sensory input of both tactile and magnetic stimuli. Therefore, the sensory modality in our system is different from the visual modality in the visual systems mentioned above.
- (2) **Biomimetic flexible 3D structure:** the sensor structure in our system employs a biomimetic, flexible, three-dimensional (3D) structure. This innovative structure is achieved through low-cost, scalable fabrication techniques in flexible electronics, allowing the entire sensor prepared in portable and wearable forms.
- (3) **Diverse sensing functions:** our system's sensing functions extend beyond visual perception, encompassing vibrotactile and magneto perception, spatiotemporal recognition, and sensory memory. Besides, our system adopts the receptor-neuron architecture and employs the fast adapting (FA) and slowly adapting (SA) encoding strategy of mechanoreceptors, imitating the neural pathway and neuronal coding of insect antennae.
- (4) **Advanced sensory processing:** sensory processing of the sensor signal is achieved using ion-gated synaptic transistors, which are specialized for synaptic processing of tactile and magnetic stimuli with static or dynamic characteristics.
- (5) **Wide range of applications:** our system enables a wide range of applications, including profile identification, Braille recognition, surface discrimination, material classification, magnetic

navigation, and touchless interfacing. These applications are important for achieving tactile intelligence and perceptual augmentation in sensory robotics and smart interfaces.

In summary, our neuromorphic antennal sensory system stands apart from currently reported artificial visual sensory systems in the aspect of sensory modality, sensor structures, sensing functionalities, neuronal functions, sensory processing, and potential applications. The comprehensive benchmarking against state-of-the-art artificial sensory systems (vision) is presented in the table below for reference.

Table S6. Benchmarking our neuromorphic antennal sensory system against the state-of-the-art artificial sensory systems (visual) in the aspect of bio-inspired and insect-inspired perception.

Bio-inspired / insect-inspired sensory system	Sensory modality	Structural characteristics	Functional characteristics	Sensory processing	Applications
Bioinspired vision sensor array with optoelectronic graded neurons [S26]	Visual (inspired by flying insect)	Planar structure on Si (fabricated by CVD, lithography)	Motion perception, spatiotemporal information encoding/fusing, temporal summation	Photo-transistor (device-level)	Motion detection, movement recognition
Memristor-based biomimetic compound eye [S27]	Visual (inspired by locust's compound eye)	Hemispherical structure on PDMS (fabricated by sputtering, spin coating)	Wide field-of-view (FoV) detection, looming detection	Memristor (device-level)	Machine vision, collision avoidance
Bioinspired vision sensors [S28]	Visual (inspired by retina)	Planar structure on Si (fabricated by CVD, lithography)	Scotopic / photopic adaptation	Photo-transistor (device-level)	Image recognition
Vertically integrated spiking cone photoreceptor array [S29]	Visual (inspired by drosophila's cone photoreceptor)	Planar structure on Si (fabricated by sputtering)	Spike encoding, light sensing	Memristor (device-level)	Color perception
Biomimetic eye with a hemispherical perovskite nanowire array retina [S30]	Visual (inspired by eye)	Hemispherical structure on PDMS (fabricated by electro-deposition, etching)	Image-sensing	Computer	Machine vision
Amphibious artificial vision system with a panoramic visual field [S31]	Visual (inspired by crab's compound eye)	Spherical structure on plastic (fabricated by molding, laser ablation, lithography, dry etching)	Amphibious imaging, panoramic imaging	Computer	Panoramic motion detection, obstacle avoidance
Artificial flexible visual memory system [S32]	Visual (inspired by retina)	Planar structure on plastic (fabricated by direct printing, deposition)	Visual memory, light detection	Memristor (device-level)	Image processing
Neuromorphic antennal sensory system (this work)	Tactile and magnetic (inspired by insect antennae)	3D structure on plastic (fabricated by solution processing, laser patterning)	Vibrotactile and magneto perception, spatiotemporal recognition, sensory memory	Synaptic transistor (device-level)	Profile detection, texture detection, robotic navigation, touchless interface, material classification

The reviewer may refer to the following revised content in the supplementary information:

- Supplementary Table S6.
- Supplementary Reference S26-S32. (bioinspired visual systems)
- Supplementary Reference S15-S25. (bioinspired flexible sensors)
- Supplementary Note 6. Comparison with Existing Bioinspired Flexible Sensors
- Supplementary Note 12. Comparison with bio-inspired visual sensory systems

The reviewer may also refer to the following revised content in the manuscript:

- “In contrast to existing bioinspired tactile sensors, our sensor displays distinct characteristics in both sensor structure and function, as well as signal encoding and processing.”
- “Distinguishing itself from state-of-the-art artificial sensory systems (both tactile and visual), our system exhibits unique features in system architecture, sensor structure, signal processing, sensory functions, and neuronal characteristics (Table S5, Table S6, Supplementary Note 12).”

Reviewer #2

Summary Comment from Reviewer #2:

In the manuscript entitled “neuromorphic electronic antennae”, the authors introduce the design of insect-inspired artificial sensory systems. Spatiotemporal sensory encoding and recognition are achieved, and vibrotactile and magneto perception capabilities are demonstrated. This work is interesting, and it may inspire the field of bioinspired electronics and neuromorphic intelligence. There are several minor issues that need to be resolved before acceptance of the manuscript:

Response:

We thank the reviewer for providing valuable insights during the evaluation of our work. In response to the reviewer's feedback, we have incorporated detailed explanations on the discrimination of tactile and magnetic sensations, the sensing mechanism of the sensor, the functionalities of the synaptic transistor, and the relevant terminology and concepts within the field of neuroscience. Furthermore, we provide a comprehensive comparison of our work with other insect-inspired visual sensory systems. Furthermore, we have enhanced the language of the manuscript to improve clarity and readability. Extensive revisions have been made to both the manuscript and the supporting information, with major modifications highlighted in RED. The detailed responses to the reviewer's comments are presented below in a point-by-point manner.

Comment 1 from Reviewer #2:

In figure 2f and figure 2i, how to differentiate between object tactile contact and non-contact magnetic proximity?

Response:

We thank the reviewer for this insightful comment.

As shown in Figure 2f below, during the process of tactile contact, the sensor is operated in contact mode. The contact with the non-magnetic object leads to the downward bending of the flexible sensor, and the subsequent retraction of the object results in the mechanical recovery of the flexible sensor. Thus, the sensor's response shows a sudden decrease followed by an increase. In contrast, as depicted in Figure 2i, the process of magnetic proximity performed in non-contact mode induces different behavior of the sensor. Specifically, the approach and withdrawal of the magnetic object cause the upward bending and subsequent mechanical recovery of the flexible sensor, and hence the sensor's response shows a gradual increase followed by a decrease.

Therefore, it is evident that the directional evolution of the sensor response differs between contact-mode tactile sensation and non-contact magnetic sensation. Such distinction underscores the sensor's capabilities in discerning the modality of sensory stimuli.

Figure 2. (f) Detection of static tactile stimuli in contact mode under different deflections of the artificial antennae. (i) Detection of magnetic proximity in non-contact mode under different sensor-to-object distances.

The reviewer may refer to the following original content in the manuscript:

- “During the approaching and retreating process of the magnetic source in a contactless manner, the sensor signal reveals that the artificial antennae underwent upward deflection followed by downward recovery (Fig. 2i), which differs from the case of contact-mode tactile sensation (Fig. 2f).”

Comment 2 from Reviewer #2:

More explanation on the sensing mechanism of the sensor is needed. Is the sensor capable of identifying the bending direction?

Response:

We appreciate the valuable suggestion provided by the reviewer.

The sensing mechanism of the sensor is based on the piezoelectric effect and magnetic interaction. The first figure below illustrates the working mechanism for tactile sensing. The sensing material in our

flexible sensor is piezoelectric materials (PVDF), and this material can generate electric charge in response to applied mechanical force or stress. This electromechanical energy conversion is the piezoelectric effect. In the absence of external force, the piezoelectric film of the flexible sensor does not deform and no electric signal is generated. Upon the application of an external tactile force (denoted as F_{tactile}), the flexible sensor is bent, inducing lateral strain of the piezoelectric material and a decrease of its polarization. This deformation triggers a potential difference (built-in electric field), and thus an electric current is generated ($I > 0$). When the bending-induced strain is released, an electric current with the opposite direction ($I < 0$) is generated.

Working mechanism of the sensor for tactile sensing.

The second figure below illustrates the working mechanism for magnetic sensing. Our flexible sensor can be modeled as a flexible beam loaded with a magnetic appendage (NdFeB) that is axially magnetized. In the absence of magnetic field, the flexible sensor is not deformed. When a magnetic or ferromagnetic object approaches to the flexible sensor, the interaction between the magnetic appendage and the object is enhanced, manifested by an increasing magnetic interaction force (denoted as F_{magnetic}). Consequently, the flexible sensor is deformed due to the magnetic interaction force, and the piezoelectric material attached to the flexible film generates an electric current ($I < 0$) during this deformation. Upon the retraction of the magnetic/ferromagnetic object, the magnetic interaction force is reduced and the flexible sensor is gradually recovered to its initial state. This recovery process results in an opposite current ($I > 0$) compared to the bending process.

Working mechanism of the sensor for magnetic sensing.

Identification of bending direction during tactile sensation can be achieved by evaluating the changing direction of the sensor's output signal. The electric charge generated by the sensor is converted into a voltage signal using our spike-encode circuit, and thus the output signal of the sensor is in the form of voltage signal. As demonstrated in Figure 2f and Figure 2i, a downward bending of the flexible sensor corresponds to a decrease in the output voltage signal, while an upward bending of the flexible sensor results in an increase in the output voltage signal. Therefore, the sensor demonstrates the capability to discern the bending direction.

The reviewer may refer to the following revised content in the manuscript:

- “The design of our electronic-antennae sensor follows the working principle of ant antennae (Fig. 2b). A pair of antennae structures were created by bending two antennae patterns on a laser-cut plastic substrate using socket-like ramps (Fig. S2), which allow the top segment of the artificial antennae to deflect upon contact while keeping the bottom segment fixed. A piezoelectric film (metalized surface) was attached to the surface of the artificial antennae (Fig. S2). Tactile force-induced bending of the sensor generated a piezoelectric signal, providing tactile-sensation functionality (Supplementary Note 2). Additionally, a magnetic appendage (uniaxially magnetized) was installed on the tip of the artificial antennae (Fig. S2). The presence of a magnetic or ferromagnetic object induced deformation of the artificial antennae through magnetic interaction, thereby enabling magneto-sensation functionality (Supplementary Note 2).”

The reviewer may refer to the following revised content in the supplementary information:

- Supplementary Note 2. Sensing mechanism of the electronic-antennae sensor

Comment 3 from Reviewer #2:

Please elaborate on the significance and functionality of using synaptic transistors for sensory recognition, instead of using other methods such as machine-learning algorithm or Fourier signal analysis.

Response:

We thank the reviewer for this thoughtful comment.

In our research, the utilization of synaptic transistors for synaptic sensory processing presents important functionalities and holds great significance for several reasons:

(1) Device-level implementation: sensory processing utilizing synaptic transistors is a hardware-based method implemented in neuromorphic devices, eliminating the need for extensive computational and memory resources. In contrast, software-based methods, such as machine learning and Fourier signal analysis, require the execution of algorithms on a central processing unit (CPU) or digital signal processor (DSP).

(2) Efficient parallel processing: multi-terminal synaptic transistors, designed to emulate biological neuron networks, excel in processing spatiotemporal patterns of sensory stimuli in an efficient and parallel manner. This enables the implementation of advanced sensory functions, including spatiotemporal recognition, sensory memory, and multisensory integration.

(3) Spike-based sensory coding: the signal for synaptic transistor is encoded as slowly-adapting and fast-adapting spikes, emulating the sensory coding strategy of biological mechanoreceptors. The neuromorphic sensory system, which implements spike-based sensory coding and synaptic sensory processing, exhibits lower circuit complexity and energy consumption compared to the complementary metal-oxide semiconductor (CMOS) based sensory system, attributed to its pulse-driven, energy-efficient operations.

The reviewer may refer to the following content (originally appeared) in the manuscript:

- “The ionic-gated synaptic transistor provides a hardware platform for emulating the synaptic functions of sensory neurons.”
- “These device characteristics lay the foundation for recognizing the spatiotemporal patterns of multiple sensory inputs.”
- “The system follows the labeled-line model of the receptor-neuron pathway, where SA and FA spike trains are separately sent to two synaptic transistors functioning as specialized neurons. This enables event-based, highly efficient, parallel processing of spatiotemporal patterns of sensory stimuli.”

Comment 4 from Reviewer #2:

The knowledge or background of neuroscience should be explained to help the readers better understand the concepts. For instance, fast-adapting and slowly-adapting spikes in mechanoreceptors.

Response:

We thank the reviewer for this suggestion, as it may improve the readability of the manuscript.

In response, we have provided detailed explanations for specific neuroscience terminology and concepts, and the aim is to elucidate key characteristics related to fast-adapting and slowly-adapting spikes in mechanoreceptors, as well as the concept of sensory cues.

(1) Fast-adapting and slowly-adapting spikes: Different mechanoreceptors generate various types of neuronal spikes. Fast-adapting (or rapidly adapting) mechanoreceptor adapts rapidly to changes in stimuli (such as vibrations), producing transient responses. Fast-adapting spikes are generated by fast-adapting mechanoreceptors during the start and end of dynamic stimulation. Fast-adapting mechanoreceptors that quickly adapt and return to a normal pulse rate have "phasic" firing patterns. The Pacinian corpuscle receptor is a classic example of fast-adapting type receptors. Slowly-adapting

mechanoreceptor produces sustained responses to static stimulation (such as pressure). Slowly-adapting spikes are generated by slowly-adapting mechanoreceptor during the entire duration when a static stimulus is presented. Slowly-adapting mechanoreceptors that are slow to return to their normal firing rate exhibit "tonic" firing patterns. The Ruffini nerve ending is an example of slowly-adapting type receptors. In general, fast-adapting mechanoreceptors with phasic firing spikes are useful in sensation of textures or vibrations, while slowly-adapting mechanoreceptors with tonic firing spikes are critical for proprioception and pressure detection.

(2) Sensory cue (sensory input): A sensory cue is the signal extracted from the sensory input by a sensory system, contributing to the process of perception. Representative examples of sensory cues include visual cues, auditory cues, and tactile cues, each corresponding to distinct sensory modalities. These sensory cues provide independent sources of sensory information, and the brain integrates a multitude of sensory cues (i.e., cross-modal and multi-modal sensory integration) to make perceptual inferences and guide motor behaviors.

In consideration of these explanations, we have revised the manuscript to incorporate these concepts and we believe that such modification will contribute to a clearer presentation of the neuroscience-related terminology.

The reviewer may refer to the following revised content in the manuscript:

- “This system adopts the connection architecture of receptors and neurons and also employs the encoding strategy of fast-adapting (FA; sensitive to dynamic stimulation) and slowly-adapting (SA; sensitive to static stimulation) mechanoreceptors to imitate the neural pathway and neuronal coding observed in biological antennae.”
- “The antennae are replete with diverse sensory receptors and neurons, comprising what is known as Johnston's organ. These components serve to identify and transcode movements of the antennae into slowly-adapting (tonic firing for sustained stimuli) and fast-adapting (phasic firing for changing stimuli) neural spikes (Fig. 1b).”
- “Capitalizing on the highly adaptive structures and sensorimotor functions of their antennae, ants demonstrate an acute ability to discern the texture and shape of their surroundings through antennal movement, akin to the whisking behaviors observed in rats (Fig. 4a). Remarkably, these perceptive abilities persist even in the absence of light, circumventing the need for sensory input from the visual system.”

Comment 5 from Reviewer #2:

The authors are suggested to compared their works with other recent insect-inspired research works (Nature Nanotechnology, 2023, 18, 882-888).

Response:

We thank the reviewer for providing this useful suggestion, which we believe will enhance the manuscript by emphasizing the uniqueness of our work.

In response, we have incorporated a comprehensive comparison with several recently reported works on insect-inspired and bio-inspired artificial sensory systems. This comparative analysis highlights the distinctive features of our neuromorphic antennal sensory system in contrast to state-of-the-art artificial visual sensory systems. The following representative works were selected for comparison:

- Optoelectronic graded neurons for bioinspired in-sensor motion perception. *Nature Nanotechnology* **18**, 882–888 (2023).
- Memristor-based biomimetic compound eye for real-time collision detection. *Nature Communications* **12**, 5979 (2021).
- Bioinspired in-sensor visual adaptation for accurate perception. *Nature Electronics* **5**, 84–91 (2022).
- Vertically integrated spiking cone photoreceptor arrays for color perception. *Nature Communications* **14**, 3444 (2023).
- A biomimetic eye with a hemispherical perovskite nanowire array retina. *Nature* **581**, 278–282 (2020).
- An amphibious artificial vision system with a panoramic visual field. *Nature Electronics* **5**, 452–459 (2022).
- An artificial flexible visual memory system based on an UV-motivated memristor. *Advanced Materials* **30**, 1705400 (2018).

In general, most of the state-of-the-art artificial sensory systems are visual sensory systems inspired by mammalian eyes or the compound eyes of insects, where the sensory input predominantly consists of visual signals. Besides, these bioinspired visual sensory systems utilize planar structures on rigid silicon or spherical structures on flexible substrates, and such designs lack three-dimensional structures with flexible or adaptive attributes. Moreover, the fabrications of the sensor and the device in these systems require chemical or physical deposition and other complicated manufacturing procedures to grow functional materials, thereby incurring high fabrication costs. Emerging artificial visual sensory systems, employing in-sensor or near-sensor computing architectures, can incorporate both the sensing and processing functionalities and simplify the layout and fabrication. Nevertheless, these systems still rely on silicon fabrication and packaging methods for constructing integrated sensory-computing architectures with interconnections. In the aspect of sensing functionalities, bioinspired visual systems can achieve various functions of visual perception, such as motion and movement perception (inspired by the fly's compound eye), panoramic and amphibious imaging (inspired by the crab's compound eye), wide field-of-view detection (inspired by the locust's compound eye), and scotopic/photopic adaptation (inspired by the retina). Some of the bioinspired visual sensory systems implement visual perception functionalities at the device level using memristors or phototransistor arrays to enable sensory processing or sensory memory. Potential applications of these systems predominantly lie in machine vision or computer vision, spanning motion detection, movement recognition, image recognition, and color perception.

In contrast, our neuromorphic antennal sensory system exhibits several important features that are different from the recently reported artificial visual sensory systems. First, our system is inspired by the sensory organ of the insect's antennae, and the sensory input is multimodal signals of tactile and magnetic

stimuli, diverging from the visual input of its counterparts. Second, the sensor structure in our system adopts a biomimetic, flexible, three-dimensional (3D) structure, and the sensor can be obtained by low-cost, scalable fabrication techniques in flexible electronics. This characteristic enables the creation of the entire system in portable and wearable forms. Third, our system's sensing functions extend beyond visual perception, encompassing vibrotactile and magneto perception, spatiotemporal recognition, and sensory memory. Besides, our system adopts the receptor-neuron architecture and employs the fast adapting (FA) and slowly adapting (SA) encoding strategy of mechanoreceptors, imitating the neural pathway and neuronal coding of insect antennae. Fourth, sensory processing in our system is achieved by using ion-gated synaptic transistors, which are specialized for the synaptic processing of tactile and magnetic stimuli with static or dynamic characteristics. Fifth, our system opens up a broad range of applications, including profile identification, Braille recognition, surface discrimination, material classification, magnetic navigation, and touchless interfacing. These applications play a pivotal role in realizing tactile intelligence and perceptual augmentation in sensory robotics and smart interfaces.

In summary, our neuromorphic antennal sensory system differs from the currently reported artificial visual sensory systems in the aspect of sensory modality, sensor structures, sensing functionalities, neuronal functions, sensory processing, and potential applications. We benchmark our neuromorphic antennal sensory system against state-of-the-art artificial sensory systems (particularly those related to visual perception), as shown in the table below.

Table S6. Benchmarking our neuromorphic antennal sensory system against the state-of-the-art artificial sensory systems (visual) in the aspect of bio-inspired and insect-inspired perception.

Bio-inspired / insect-inspired sensory system	Sensory modality	Structural characteristics	Functional characteristics	Sensory processing	Applications
Bioinspired vision sensor array with optoelectronic graded neurons [S26]	Visual (inspired by flying insect)	Planar structure on Si (fabricated by CVD, lithography)	Motion perception, spatiotemporal information encoding/fusing, temporal summation	Photo-transistor (device-level)	Motion detection, movement recognition
Memristor-based biomimetic compound eye [S27]	Visual (inspired by locust's compound eye)	Hemispherical structure on PDMS (fabricated by sputtering, spin coating)	Wide field-of-view (FoV) detection, looming detection	Memristor (device-level)	Machine vision, collision avoidance
Bioinspired vision sensors [S28]	Visual (inspired by retina)	Planar structure on Si (fabricated by CVD, lithography)	Scotopic / photopic adaptation	Photo-transistor (device-level)	Image recognition
Vertically integrated spiking cone photoreceptor array [S29]	Visual (inspired by drosophila's cone photoreceptor)	Planar structure on Si (fabricated by sputtering)	Spike encoding, light sensing	Memristor (device-level)	Color perception
Biomimetic eye with a hemispherical perovskite nanowire array retina [S30]	Visual (inspired by eye)	Hemispherical structure on PDMS (fabricated by electro-deposition, etching)	Image-sensing	Computer	Machine vision
Amphibious artificial vision system with a panoramic visual field [S31]	Visual (inspired by crab's compound eye)	Spherical structure on plastic (fabricated by molding, laser ablation, lithography, dry etching)	Amphibious imaging, panoramic imaging	Computer	Panoramic motion detection, obstacle avoidance
Artificial flexible visual memory system [S32]	Visual (inspired by retina)	Planar structure on plastic (fabricated by direct printing, deposition)	Visual memory, light detection	Memristor (device-level)	Image processing
Neuromorphic antennal sensory system (this work)	Tactile and magnetic (inspired by insect antennae)	3D structure on plastic (fabricated by solution processing, laser patterning)	Vibrotactile and magneto perception, spatiotemporal recognition, sensory memory	Synaptic transistor (device-level)	Profile detection, texture detection, robotic navigation, touchless interface, material classification

The reviewer may refer to the following revised content in the manuscript and supplementary information:

- Reference 45.
- Supplementary Table S6.
- Supplementary Reference S26-S32.
- Supplementary Note 12. Comparison with bio-inspired visual sensory systems

Reviewer #3

Summary Comment from Reviewer #3:

In this paper, the authors reported an electronic antennal nerve system comprising electronic antennae sensors and artificial synaptic devices inspired by an ants' antenna. The electronic antennae sensor, constructed from piezoelectric PVDF film, facilitates signal transmission through the integration of a coding circuit. Additionally, the encoded signals serve as input for integrated artificial synaptic devices based on ion gel. Based on this process, the electronic antennal nerve system demonstrates capabilities in vibrotactile and magneto perception. While the results are intriguing, certain aspects warrant further clarification.

Response:

We express our sincere gratitude to the reviewer for their invaluable comments. In addressing the detailed feedback, we have enhanced the manuscript by providing detailed explanations highlighting the unique advantages and characteristics of our neuromorphic antennal sensory system, comparing it with existing sensors and devices reported in the literature. Moreover, we have incorporated experimental results on device-to-device variations and also developed the modulation strategy of device memory to enhance recognition accuracy. Specific justifications for selecting fast-adapting (FA) or slow-adapting (SA) spikes in tasks related to tactile and magneto perception have been included. Additionally, we have explored the impact of movement speed on the output signal during surface scanning and demonstrated experimental strategies to improve recognition accuracy for profile detection and texture classification. Further refinement has been applied to the language of the manuscript to improve overall clarity and readability. Both the manuscript and supporting information have undergone extensive revisions, with the major modifications highlighted in RED. Detailed point-by-point responses to the reviewer's comments are presented below.

Comment 1 from Reviewer #3:

There have been several reports on sensors using PVDF and synaptic devices utilizing ion gel (Journal of Materials Chemistry C 9, 8372-8394 (2021), ACS Applied Nano Materials 6, 1522-1540 (2023)). It would enhance the paper's informativeness to clarify the unique perspectives introduced beyond the combination of these devices into a neuromorphic electronic antenna.

Response:

We thank the reviewer for this insightful comment on PVDF sensors and synaptic devices.

In addressing the reviewer's comment, we conducted a thorough review of recently published articles in the field, focusing on PVDF-based sensors and ion-gel synaptic devices. The literature we examined include the two reference articles specifically mentioned by the reviewer, along with other relevant works:

- A review on PVDF nanofibers in textiles for flexible piezoelectric sensors. *ACS Applied Nano Materials* 6, 1522-1540 (2023).
- Recent progress in artificial synaptic devices: materials, processing and applications. *Journal of Materials Chemistry C* 9, 8372-8394 (2021).
- Piezoelectric biomaterials for sensors and actuators. *Advanced Materials* 31, 1802084 (2019).
- Emerging trends in soft electronics: integrating machine intelligence with soft acoustic/vibration sensors. *Advanced Materials* 35, 2209673 (2023).
- Flexible PVDF based piezoelectric nanogenerators. *Nano Energy* 78, 105251 (2020).
- Recent advanced applications of ion-gel in ionic-gated transistor. *Npj Flexible Electronics* 5, 13 (2021).
- A comprehensive review on emerging artificial neuromorphic devices. *Applied Physics Reviews* 7, 011312 (2020).

In these literature articles, the PVDF-based sensors mostly adopt multilayer planar structures composed of films, yarns, filaments, or fibers, lacking the three-dimensional biomimetic structure demonstrated in our neuromorphic antennal sensory system. Besides, existing PVDF-based sensors typically serve unimodal sensing functions, such as pressure, motion, or vibration sensing. The recently reported ion-gating synaptic devices mostly utilize dielectric materials like ionic liquid (such as EMIM-TFSI) or chitosan gel to prepare the ion-gel or ionic-liquid layer. The semiconducting material in these devices is predominantly confined to single component, such as metal oxide, transition-metal dichalcogenide, or organic material, organized in multilayer or heterojunction structures, while mixed-dimension material, such as the combination of thin film, 1D nanowires, or 2D nanoflakes, are scarcely reported. Moreover, hardware connections of these synaptic devices in the literature simply employ cross-bar or linear array architectures, without considering the neuronal encoding and neuronal model in biological synapse networks.

By comparing with the reported works on PVDF-based sensors and ion-gel synaptic devices, our neuromorphic antennal sensory system exhibits unique characteristics and advantages:

(1) System Integration: our system realizes a 2-by-2 receptor-neuron network by emulating the “labeled-line” model of the biological receptor-neuron pathway. This approach differs from directly connecting the PVDF sensors to the synaptic devices. Utilizing different encoding strategies (fast-adapting and slowly-adapting) for the sensor signals enables the separate processing of static and dynamic signals. Device-level cognitive perception of spatiotemporal sensory information was further achieved using neuromorphic hardware in a parallel, pulse-driven, energy-efficient manner, making our system different from currently reported artificial sensory systems relying on external computational resources.

(2) Structure and Function of Sensor: our PVDF-based electronic antennae sensor features segmented, flexible, three-dimensional structures and multifunctional capabilities of vibrotactile and magnetic perception. Our sensor essentially emulates both the structure and functions of ant antennae, breaking the design constraints of conventional PVDF sensors with tactile function and planar structure. Furthermore, our sensor can be operated in various modes to detect stimuli such as contact, vibration, and magnetic field with very low detection limit. This enables potential applications,

including profile detection, texture discrimination, Braille code recognition, material classification, magnetic navigation, and touchless interfacing. These applications are challenging to achieve using conventional PVDF sensor.

(3) Material and Function of Synaptic Device: our flexible synaptic transistor is fabricated using nanoflake-adsorbed metal oxide thin film (mixed-dimension material) and alginate-based ion gel (naturally occurring material). These materials can promote the charge transfer dynamics and the ionic gating effect, contributing to the optimization of synaptic and memory properties of the device. Importantly, we introduce a novel strategy (application of sustaining spikes) for regulating the sensory memory of the device, crucial for application-specific sensory processing with different memory requirements. Compared with recently reported synaptic devices using ion gel, our device exhibits unique synaptic and neuronal functions, such as spatiotemporal integration, multimodal sensory processing, and controllable sensory memory. These characteristics are useful for vibrotactile and magneto perception.

The reviewer may refer to the following revised content in the supplementary information:

- Supplementary Reference S33-S39.
- Supplementary Note 6. Comparison with Existing Bioinspired Flexible Sensors

The reviewer may refer to the following revised content in the manuscript:

- “In contrast to existing bioinspired tactile sensors, our sensor displays distinct characteristics in both sensor structure and function, as well as signal encoding and processing (Supplementary Note 6).”
- “In comparison with recently reported synaptic devices fabricated using solution-processable materials (Table S1), our device demonstrates advantages in synaptic and neuronal functions (demonstrated in the following part).”
- “Our system achieves device-level cognitive perception of sensory information in a multimodal, parallel, pulse-driven manner, distinguishing it from current artificial sensory systems that rely on external computational resources.”
- “Distinguishing itself from state-of-the-art artificial sensory systems (both tactile and visual), our system exhibits unique features in system architecture, sensor structure, signal processing, sensory functions, and neuronal characteristics (Table S5, Table S6, Supplementary Note 12).^{6,8,12,13,15,43,44,45}”

Comment 2 from Reviewer #3:

In Figure S6, the authors deposited nanoflakes on the channel via the dip-coating method. However, atomic force microscopy images reveal random deposition, potentially leading to high device-to-device variations. Thus, the high device-to-device variation characteristics will likely cause high errors during

classifications. The authors should investigate the device-to-device variation of the devices and its effect on classification accuracy.

Response:

We thank the reviewer for providing this valuable insight. The concerns raised in the reviewer's comment have been addressed as follows:

Device-to-Device Variation: First of all, as suggested by the reviewer, device-to-device variation was evaluated. A total of 12 synaptic transistors, fabricated on a flexible substrate, were tested by applying voltage spikes (+5V sensory spikes followed by +1V sustaining spikes) and recording the synaptic current. The crucial parameter of memory retention behavior, essential for sensory processing, was calculated here as the ratio of synaptic current at $t = 20$ s (acquired after the stop of spike stimulation) to the resting current (acquired before spike stimulation). Figure S14 (a) illustrates the device-to-device variation of memory retention behaviors for all 12 devices, indicating a variation of approximately 7.1%.

Analysis of Contributing Factors: Secondly, contributing factors leading to device-to-device variations are analyzed. As mentioned by the reviewer, the atomic force microscopy images (Figure S9) reveal the random distribution of nanoflakes deposited by liquid-phase adsorption. However, it should be noted that these nanoflakes are located on top of a continuous metal-oxide thin film. KPFM characterization results and the control device (Figure S9, Figure S10) without nanoflakes confirmed that the metal oxide film dominantly influences charge transport behavior, while the nanoflakes affect the current level and on/off ratio. Additionally, the electronic properties of the ion gel contribute to the synaptic response of the device. Thus, the device-to-device variation can be attributed to multiple factors, including morphology and properties of the metal oxide thin film, distribution and properties of the nanoflakes, and interface and properties of the ion gel layer. It is emphasized that our synaptic device, prepared on a flexible substrate using solution-processable methods, inherently exhibits higher device-to-device variation than other synaptic devices prepared on silicon substrates through physical or chemical deposition methods.

Mitigation Strategy: Furthermore, we developed an experimental strategy to effectively mitigate the device-to-device variations. As shown in Figure 3h, we have verified that the memory retention behavior and sensory memory of the synaptic device can be modulated by applying sustaining spikes with small amplitude (ranging from 0 to 1.5 V) after the stimuli of sensory spikes (5 V). For precise regulation of memory retention behavior, here we finely adjust the amplitude of the sustaining spike using a small step size (minimum step size 0.05V) while recording the synaptic current from the device. The results shown in Figure S14 (b) reveal that multi-level states of the synaptic current can be achieved by finely tuning the amplitude of the sustaining spikes. Therefore, memory retention behavior for each synaptic device can be individually regulated by appropriately setting the sustaining spike to ensure similar retention behavior. To implement this strategy, we experimentally modulated the memory retention behaviors of all 12 devices by adjusting the amplitude of the sustaining spike (minimum step size 0.05V), and the device-to-device variation is evaluated again. As presented in Figure S14 (c), the value of device-to-device variation is reduced from 7.1% to 2.6%. This strategy of adjusting the amplitude of sustaining spike effectively mitigates the issue of device-to-device variation by enabling manual control of device memory.

Influence on Classification Accuracy: Finally, we investigate the influence of the device-to-device variation on classification accuracy. In our original setup, as shown in Figure 4e, the classification task of

chess profile detection was executed using identical synaptic devices, yielding a recognition accuracy of 0.916, and this case did not consider the device-to-device variation. Taking account of the device-to-device variation, the classification task was repeated using different synaptic devices. The strategy of adjusting the sustaining spike as mentioned above was employed to ensure similar memory retention behaviors across diverse devices. As a result, the recognition accuracy for chess profile detection, while considering the device-to-device variation (2.6%), slightly decreased from 0.916 to 0.892. Therefore, the device-to-device variation of the device memory may diminish the classification accuracy of tactile recognition. Nevertheless, the strategy of adjusting the sustaining spikes we developed here can mitigate such influence by reducing the overall device-to-device variation.

Figure S14. Device-to-device variations of the artificial synaptic device. (a) Device-to-device variation of 12 devices evaluated by measuring their memory retention behaviors ($t = 20$ s) before implementing the strategy of adjusting sustaining spikes (+5 V sensory spikes followed by sustaining spikes with fixed amplitude of +1 V). (b) Implementation of the strategy of adjusting sustaining spikes showing that the memory retention behavior of the device can be finely regulated. (c) Device-to-device variation of 12 devices evaluated by measuring their memory retention behaviors ($t = 20$ s) after implementing the strategy of adjusting sustaining spikes (+5 V sensory spikes followed by sustaining spikes with varied amplitude).

The reviewer may refer to the following revised content in the supplementary information:

- Supplementary Figure S14.
- Supplementary Note 7. Evaluation and improvement of device-to-device variations

The reviewer may refer to the following revised content in the manuscript:

- “This strategy enables the regulation of sensory memory in the device by adjusting the voltage of the sustaining spikes, offering adaptability for application-specific sensory processing with varying memory requirements. Furthermore, employing this strategy allows the modulation of memory retention behaviors across different devices, addressing concerns related to device-to-device variations (Supplementary Note 7, Fig. S14).”

Comment 3 from Reviewer #3:

In Figure 3j, the authors stated that both fast-adapting (FA) and slow-adapting (SA) are being utilized for the classification. However, it appears that only FA was used in Figure 4f and 4i, while only SA was used in Figure 5f and 5k. A clarification is needed regarding the specific reasons for choosing either FA or SA in these instances.

Response:

We thank the reviewer for this helpful suggestion.

In response to the reviewer's comment, we have added clarifications regarding the specific reasons for choosing fast-adapting (FA) and slow-adapting (SA) spikes in various classification tasks. The detailed explanations are provided below:

In biology, fast-adapting (FA) and slowly-adapting (SA) spikes reveal different properties inherent to sensory input. Neuronal spikes characterized by FA and SA patterns are generated from different mechanoreceptors. Fast-adapting (or rapidly adapting) mechanoreceptor adapts rapidly to changes in stimuli (such as vibrations), producing transient responses. FA spikes are generated by fast-adapting mechanoreceptors during the start and end of dynamic stimulation. FA spikes exhibit "phasic" firing pattern characterized by rapid adaptation and prompt return to baseline pulse rates. Conversely, slowly-adapting (SA) mechanoreceptors produce sustained responses to static stimulation (such as pressure). SA spikes are generated by slowly-adapting mechanoreceptors throughout the entire duration when a static stimulus is presented. SA mechanoreceptors exhibit "tonic" firing patterns, characterized by a slow return to their normal firing rates. In terms of sensory functions, fast-adapting mechanoreceptors with FA spikes are useful in sensing textures or vibrations, while slowly-adapting mechanoreceptors with SA spikes play a pivotal role in perceiving position, movement, shape, and pressure.

Inspired by the sensory encoding mechanism observed in biological sensory systems, our work encodes sensory information into SA and FA spikes to capture static and dynamic characteristics of stimuli, respectively. SA spikes with tonic firing patterns are utilized for recognizing slow-changing sensory input with static characteristics, such as profile information and magnetic interactions. In contrast, FA spikes with phasic firing patterns are chosen for tasks involving rapid-changing sensory input with dynamic characteristics, such as vibration information. Consequently, the selection of SA or FA spikes is task-specific, ensuring reliable and accurate perception: SA spikes are employed for sensing tasks including

chess profile detection (Figure 4c) and magneto-perception (Figure 5c, Figure 5f, Figure 5k), where a slow response is essential for identifying profile information or interaction force. FA spikes are utilized for sensing tasks including surface pattern detection (Figure 4f) and material texture discrimination (Figure 4i), where a fast response is necessary to identify the textures and patterns with high resolution. It is noteworthy that, at the system-level, our system can simultaneously address SA and FA spikes, both encoded from the sensor signal and selectively processed in the synaptic devices. As illustrated in the original manuscript, our system design (Figure 3j), based on the receptor-neuron model, allows quantification of static and dynamic characteristics of spatiotemporal stimuli through synaptic currents recorded from the SA and FA devices. For accurate and reliable recognition of sensory signals, one type of sensory spike is chosen, leading to a biased sensory processing. This strategy is analogous to the perceptual weighing function observed in the brain, where the biased assignment of perceptual weights to specific sensory stimuli is contingent on the reliability and intensity of sensory inputs.

In summary, slow-adapting (SA) and fast-adapting (FA) sensory spikes correspond to the static and dynamic properties of the sensory input. Both SA and FA spikes are sensory spikes carrying the spatiotemporal information of the sensory stimuli. At system level, our neuromorphic system can simultaneously process these two types of sensory spikes. For a specific perception task, the appropriate spike type is chosen based on the changing behavior of the sensory stimuli, enhancing the reliability and accuracy of the perceptual recognition.

The reviewer may refer to the following revised content in the supplementary information:

- **Supplementary Note 15. Fast-adapting (FA) and slowly-adapting (SA) spikes**

The reviewer may refer to the following content in the manuscript:

- “Given that the SA spikes with static characteristics better reflect the profile information, the SA device’s synaptic currents for all six chess pieces were compared (Fig. 4d), revealing variations across all cases in the ending value of the SA device’s output recorded at the end of a sensory event.”
- “Further experiments were performed to investigate the mechano-sensation performance in discerning delicate patterns and textures. Given that the FA spikes with dynamic characteristics better represent the vibration information, the output of FA device was utilized for recognition tasks.”
- “Magnetic stimuli were essentially transformed into the gradual deflection of the artificial antennae due to the magnetically responsive characteristic of the sensor. The SA spikes characterized by slowly changing behaviors were utilized for magneto-perception.”
- “Various materials produced sensor signals with diverse profiles (Fig. 5e), and the encoded SA spikes derived from the sensor signal (Fig. 5f) elucidate the magnetic interaction process, resulting in distinct synaptic currents within the SA device (Fig. 5g). Consequently, material classification tasks can be successfully executed by analyzing the output of the SA device (Fig. 5h).”

- “Again, the SA spikes reveal the slowly changing sensory information of magnetic interaction, and the types of finger motion can be classified based on the ending value of the SA device’s synaptic current (Fig. 5l, Fig. S26).”

Comment 4 from Reviewer #3:

In Figure 4b, the authors analyzed the surface of the object by moving the device at a constant speed. In that case, the frequency or amplitude of the signal may vary with the movement speed of the device. Thus, the movement speed can affect the total accuracy of the classification, which should be further analyzed. It would be valuable if the authors to show the effect of movement speed on the output signals and their corresponding accuracy.

Response:

We express our gratitude to the reviewer for the insightful comment regarding the movement speed of the sensor during tactile perception.

In response to this suggestion, we have performed additional experiments to systematically investigate the influence of movement speed on the frequency and amplitude of the sensor signal. Surface patterns with periodic ridges were used as the object of interest for tactile perception. A total of four surface patterns with different ridge widths (0.5, 0.7, 0.9, 1.1 mm) were laterally scanned by the flexible sensor at various movement speeds (2, 4, 6, 8 mm s⁻¹).

Analysis of Sensor Signal: The obtained sensor signals are presented in Figure S20 (a)-(d). It can be observed from the experimental results that as the scanning speed increases, the frequency of the sensor signal increases correspondingly, while the intensity (amplitude) of the signal does not change obviously. A closer look at the sensor signal further reveals that lateral sliding at different movement speeds causes the sensor to vibrate significantly, with each stripe of the surface pattern triggering a tactile signal featuring oscillation behavior.

Analysis of Device Output: Moreover, the output of the fast-adapting (FA) device during surface scanning process is presented in Figure S20 (e), and the device output exhibits stepwise spiking, corresponding to the periodic stripes of the sample surface. The time interval between each spiking event is crucial for matching sensory memory and facilitating sensory processing. It can be thus inferred that the scanning speed, affecting the timing of spiking events in device output, needs to be controlled within a reasonable range.

Impact of Movement Speed: We further examine the effect of movement speed on the recognition accuracy in a tactile perception task. Figure S20 (f) shows the relationship between the movement speed of the sensor and the accuracy of surface pattern recognition. The classification accuracy is the highest under the movement speed of 6 mm s⁻¹, meaning that 6 mm s⁻¹ is the optimal movement speed for the surface pattern recognition task. Note that in our original setup, the movement speed of the sensor was chosen as 5 mm s⁻¹, which is close to the optimal value, and the recognition experiments we performed are reliable. A slow scanning speed will increase the time interval between tactile event, and this may cause memory loss

and reduce recognition accuracy significantly. Conversely, a fast scanning speed may cause the “shadow effect”, similar to the case when an atomic force microscopy tip scans quickly across a sample surface, and consequently the sensor signal cannot resolve the surface patterns, leading to reduced recognition accuracy. The relationship between movement speed and recognition accuracy highlights the significance of selecting an appropriate scanning speed.

Task-Specific Variation in Movement Speed: In addition, we find out that the optimal scanning speed may vary between different tactile perception tasks, since different tactile tasks may have different requirements on sensory memory and sensory processing. For instance, the profile detection task requires a relatively slower movement speed compared to the surface pattern recognition task. This variation aligns with human tactile exploration, where individuals employ different speeds for perceiving object shape and surface texture.

In conclusion, the movement speed of the sensor during tactile perception has a great impact on the frequency of the sensor signal, with minimal impact on the amplitude. The selection of an appropriate movement speed of the sensor is crucial for reliable recognition accuracy, considering its effects on sensory memory and processing. In our specific case, the movement speed of the sensor during tactile perception was appropriately set to ensure reliable tactile recognition results.

Figure S20. Influence of the scanning speed of the sensor on surface recognition performance. (a–d) Sensor signals acquired at various scanning speed of the sensor when surface patterns with ridge widths of 0.5 mm (a), 0.7 mm (b), 0.9 mm (c), and 1.1 mm (d) were laterally scanned. (e) Representative output of the FA device showing the stepwise spiking, corresponding to the periodic stripes of the sample surface. (f) Relationship between the scanning speed of the sensor and the accuracy of surface pattern recognition.

Reviewer may refer to the following revised content in the supplementary information:

- **Supplementary Note 10. Influence of scanning speed on tactile recognition**
- **Supplementary Figure S20. Influence of the scanning speed of the sensor on surface recognition performance**

Reviewer may refer to the following revised content in the manuscript:

- **“Importantly, the movement speed of the sensor during surface scanning needs to be controlled within appropriate ranges (Supplementary Note 10, Fig. S20), since it affects the surface recognition accuracy.”**
- **“Mechanoreception experiments were performed by scanning the sensor’s artificial antennae across an object’s surface at constant velocity (5 mm s^{-1}).”**

Comment 5 from Reviewer #3:

In Figure 4e, the recognition accuracy for the knight is lower compared to other chess pieces. Additionally, in Figure S21, recognition accuracy for M3 and M4 surfaces is lower than for other surfaces. The authors should provide an explanation for lower accuracy associated with specific objects or surfaces and propose potential avenues for improvement.

Response:

This reviewer's comment on enhancing recognition accuracy is highly appreciated.

In response to this comment, we have thoroughly analyzed the reasons behind reduced recognition rates in specific cases and proposed experimental methods to further improve the accuracy. The key findings and proposed methods are summarized below.

Analysis of Chess Profile Detection: For chess profile detection, the acquired sensor signal corresponds to the lateral profile of the chess, and the sensory spikes encoded from the sensor signal are indicative of the deflection and vibration behavior of the electronic antennae sensor. Given that slowly-adapting (SA) spikes with static characteristics are better suitable for reflecting profile information (Figure S15), we employ the output of the SA device and the meaning firing rate of the SA spike as classification criteria to distinguish different chess shapes. The chess recognition result (Figure 4e) reveals that the Knight is occasionally identified as the Rook, resulting in a relatively low accuracy (0.8). This reduced recognition accuracy for the Knight may be due to its similar height and curved shape compared to the Rook chess piece. Additionally, owing to the sensory memory effect, the device output and mean firing rate for the Knight are close to those of the Rook, contributing to the diminished recognition rate observed during the chess profile classification task.

Analysis of Material Texture Discrimination: For material texture discrimination, the classification criteria utilized the output of the fast-adapting (FA) device and the mean firing rate of FA

spikes. This choice was based on the fact that FA spikes with dynamic characteristics better capture vibration information (Figure 4i, Figure S21). The FA spikes briefly delineated the texture information, with the material's roughness, porosity, and periodicity affecting the spatiotemporal patterns of the sensory spikes. The texture recognition result in Figure 4e indicates reduced accuracy for M3 and M4 samples (dish sponge and canvas fabric), due to that the M3 sample is occasionally misclassified as the M4 sample. Examination of the mean firing rate of the sensory spike (Figure S22) reveals that M3 and M4 samples yield sensory spikes with close values of firing rates. Further analysis of the sample surfaces shows that both M3 and M4 samples exhibit rough surfaces with large bumps. Consequently, the similarity in surface morphology and roughness may contribute to the relatively lower recognition rate of M3 and M4 samples during the material texture discrimination task.

Methods for Improving Accuracy: To enhance the recognition accuracy for objects and textures as shown in Figure S24 (a)-(b), we propose two experimental methods. The first method is to appropriately increase the pre-deflection applied to the tip of the electronic antennae sensor during its contact with the object of interest, thereby increasing the contact depth between the sensor tip and the target object. Figure S24 (c)-(d) shows the sensor signal for chess profile detection (King) obtained at contact depth (maximum value) of 3 mm and 3.5 mm, demonstrating that an increased contact depth leads to a higher amplitude of the sensor signal (spiking intensity was enhanced). Similarly, Figure 24 (e)-(f) exhibits the sensor signal for surface texture detection (M3 and M4 samples) acquired at contact depth (maximum value) of 1 mm and 1.5 mm, revealing that an enlarged contact depth results in higher intensity and finer resolution of the surface texture information. These experimental results confirm that appropriately increasing the contact depth may improve the quality of the sensor signal, thereby potentially improving the recognition accuracy of perception tasks. It is noteworthy that increasing the contact depth will augment the contact force exerted on the sample surface, potentially causing indentation of the surface material and enlarged deflection of the sensor tip. Therefore, the contact depth should be controlled within an appropriate range (<6 mm in our setup). The second method is to employ firing rate coding for the sensory spikes. In our original setup, the sensor signal is encoded into fast-adapting (FA) and slowly-adapting (SA) sensory spikes with a fixed frequency, implying the use of a single threshold value during the encoding process. By employing firing rate coding during the encoding process, multiple threshold values can be utilized to generate sensory spikes with various frequencies, thereby increasing the efficiency of information encoding. This firing rate coding method, generating sensory spikes with rate-modulated information, holds potential to further improve the recognition accuracy of perception tasks.

Figure S24. Further improvement of tactile recognition accuracy. (a) Photograph of the six chesses used for profile detection. (b) Photograph of the six material textures (M1: metal foam, M2: patterned plastic, M3: dish sponge, M4: canvas fabric, M5: abrasive paper, M6: porous sponge) used for surface texture discrimination. (c–d) Sensor signal acquired by detecting the chess profile of “King” at contact depth of 3 mm (c) and 3.5 mm (d). (e–f) Sensor signal acquired by detecting the surface texture of M3 and M4 at contact depth of 1 mm (e) and 1.5 mm (f). The experimental results confirm that appropriately increasing the contact depth may improve the quality of the sensor signal, which may presumably improve the recognition accuracy of perception tasks.

The reviewer may refer to the following revised content in the supplementary information:

- Supplementary Note 11. Improvement of recognition accuracy
- Supplementary Figure S24. Further improvement of tactile recognition accuracy

The reviewer may refer to the following revised content in manuscript:

- “Further enhancements in recognition accuracy can be achieved through optimization of tactile contact conditions and spike encoding strategies (Supplementary Note 11, Fig. S24).”

REVIEWER COMMENTS

Reviewer #1 (Remarks to the Author):

I congratulate the authors to have join forces with a neurobiologist or alike and have considered my comments with great care, even if it meant revising substantially the MS, toning down some grandiose claims about the extend of bio-inspiration which went into your artefact, correcting some factual mistakes and adding some new materials. I think the paper is now more focused, more interesting and the real added benefit of your work made more transparent to the readers.

Reviewer #2 (Remarks to the Author):

In the revised manuscript, the authors have addressed the questions from the reviewers. I would recommend the acceptance.

Reviewer #3 (Remarks to the Author):

The revised manuscript and response letter effectively address the raised issues and questions posed by the reviewers. While some comments have been satisfactorily addressed and are well-reflected in the revised manuscript, there remain certain aspects where the explanations provided are still ambiguous. Detailed questions and comments are given below:

1. Regarding the response to comment 2, the authors thoroughly analyzed device-to-device variation and its mitigation strategy. Could the authors provide a more detailed analysis of how factors like morphology, nanoflake distribution, and ion gel properties individually influence device performance? Also, to validate the effectiveness of the mitigation strategy involving the adjustment of sustaining spike amplitude, could the authors discuss the stability and repeatability of this strategy across multiple experiments? Finally, in comparing the synaptic and neuronal functions with those reported in the literature, can the authors provide a deeper exploration of how the mixed-dimension material and alginate-based ion gel contribute to optimizing synaptic and memory properties? How do these unique features enhance the neuromorphic capabilities of the device?

2. In response to comment 3, the authors have provided clarifications on the selection of FA and SA spikes in various classification tasks. Based on the explanations provided by the authors, it seems that the adaptive selection of spikes (FA and SA) is crucial for achieving high-performance artificial antennal sensory systems. Could the authors elaborate on whether the suggested neuromorphic system can dynamically adjust its spike selection strategy in real-time based on changes in the environment or task requirements? Additionally, does the proposed system possess any learning or adaptive capabilities in terms of adjusting its spike selection strategy over time?

3. Regarding the response to comment 4, the authors noted that the optimal scanning speed for profile detection and recognition (classification) may differ. It would be helpful for the authors to elaborate on why there is a variation in the optimal scanning speed for different tactile perception tasks, such as profile detection and surface pattern recognition (classification). Since profile detection should logically precede recognition, it seems there should be no difference in optimal scanning speed between the two tasks. Also, how does the varying speed align with human tactile exploration and the specific requirements of each task?

Response to Reviewers' Comments
(Manuscript ID: NCOMMS-23-49599-A)

Reviewer #1

Summary Comments:

I congratulate the authors to have join forces with a neurobiologist or alike and have considered my comments with great care, even if it meant revising substantially the MS, toning down some grandiose claims about the extend of bio-inspiration which went into your artefact, correcting some factual mistakes and adding some new materials. I think the paper is now more focused, more interesting and the real added benefit of your work made more transparent to the readers.

Author Response:

We thank the reviewer for the appreciation of our revised work. We also express our gratitude for the time and effort the reviewer dedicated to the review.

Reviewer #2

Summary Comments:

In the revised manuscript, the authors have addressed the questions from the reviewers. I would recommend the acceptance.

Author Response:

We appreciate the reviewer's recommendation of our revised work.

Reviewer #3

Summary Comments:

The revised manuscript and response letter effectively address the raised issues and questions posed by the reviewers. While some comments have been satisfactorily addressed and are well-reflected in the revised manuscript, there remain certain aspects where the explanations provided are still ambiguous. Detailed questions and comments are given below

Author Response:

We express our sincere gratitude to the reviewer for providing these thoughtful comments and suggestions aimed at improving the quality of our manuscript. Regarding the performance of the synaptic device, we have analyzed the factors that influence device performance, discussed the stability and repeatability of the strategy of adjusting the sustaining spike's amplitude, and investigated the role of semiconducting materials and ion gel in optimizing the device's synaptic and memory properties and the reasons behind the enhanced neuromorphic capabilities. Regarding the spike selection strategy in our neuromorphic system, we have provided a feasible method for dynamic adjustment and also proposed a possible approach to achieving learning capabilities. Regarding the scanning speed of the sensor during tactile perception, we have analyzed the variations in the optimal scanning speed for different tasks, and also discussed how the task-specific varying speeds align with the observation in human tactile experiments. Additionally, some technical terms describing the tactile perception experiments have been revised to convey their meaning in a clear and understandable manner. Both the manuscript and the supporting information have been revised, with major modifications highlighted in RED. In addition, the code for spike-encoding function has been uploaded to GitHub repository (<https://github.com/Jerix1989/E-antennae.git>). The point-to-point responses to the reviewer's comments are presented below.

Comment 1 from Reviewer #3:

Regarding the response to comment 2, the authors thoroughly analyzed device-to-device variation and its mitigation strategy. Could the authors provide a more detailed analysis of how factors like morphology, nanoflake distribution, and ion gel properties individually influence device performance? Also, to validate the effectiveness of the mitigation strategy involving the adjustment of sustaining spike amplitude, could the authors discuss the stability and repeatability of this strategy across multiple experiments? Finally, in comparing the synaptic and neuronal functions with those reported in the literature, can the authors provide a deeper exploration of how the mixed-dimension material and alginate-based ion gel contribute to optimizing synaptic and memory properties? How do these unique features enhance the neuromorphic capabilities of the device?

Author Response:

We would like to thank the reviewer for providing these valuable comments.

In our first version of the response letter, we have mentioned that “the device-to-device variation can be attributed to multiple factors, including morphology and properties of the metal oxide thin film, distribution and properties of the nanoflakes, and interface and properties of the ion gel layer”. To address the reviewer comment on how these factors individually influence device performance, we added detailed analysis and explanations for each factor.

We start with explanation of the working mechanism. The operating principle of our synaptic transistor is based on ionic gating. By applying positive voltage spike trains to the planar gate, the cations (protons) in the ion-gel dielectric layer drifted and trapped on the interface between the ion gel and the semiconductor channel (Figure R1 below). This resulted in the accumulation of carriers (electrons) in the channel surface and the formation of an electric double layer (Figure R1 below) due to the electrostatic effect. The scenario reverses when negative voltage spikes are applied to the gate.

We then individually analyze the factors that influence device performance. Firstly, our previous results on material characterization and device fabrication shows that morphology of the metal-oxide semiconducting thin film is critical for achieving desired transistor properties of the device. The metal-oxide semiconducting film with too thin thickness or with too many defects will result in discontinuous or pin hole structures, which are detrimental to the formation of a uniform semiconducting channel. Our optimized fabrication process produces a smooth and continuous metal-oxide film (Figure R2 below, SEM image) by treating the substrate using UV-Ozone (10 min), which increase the surface energy and wetting capability of the substrate, and then annealing the spin-coated film at elevated temperature (300 °C for two hours), which effectively transform the precursor material into metal oxide material. These steps ensure uniform morphology and semiconducting properties of the metal-oxide film. Secondly, the distribution and properties of the nanoflakes is also important for improving the transistor and synaptic properties of the device. Note that the fabrication process, optimized for the adsorption of nanoflakes, involves purifying the nanoflake solution and controlling the conditions of the liquid-phase adsorption. This results in highly-crystalline nanoflakes (Figure R3 below, TEM image) densely distributed without obviously aggregation at the surface of metal-oxide film (Figure R4 below, SEM image). The distribution of the nanoflakes affect the device performance. When the nanoflakes are insufficiently purified, large aggregation of the nanoflakes with poor distribution will appear and reduce the gating effect (note that the 2H-phase MoS₂ nanoflakes prepared by liquid-phase exfoliation is wide band-gap semiconducting material with organic ligands, so their large aggregates at the surface of metal-oxide film will compromise the electrostatic gating). Moreover, performance comparison of the synaptic devices fabricated without and with nanoflakes (Figure R5 below) demonstrates that the nanoflakes enhance the device performance by increasing the on-off ratio (from 7.5 to 84.5), suppressing the off-state current (from 3.05 to 0.15 μA), increasing the linearity of potentiation behavior, and improving the signal-to-noise ratio of synaptic response. These enhancements can be attributed to the charge trapping effect of the nanoflakes located at the interface between semiconducting metal-oxide film and the alginate ion gel. Additionally, shifts in the threshold voltage (from 1.0 to 2.4 V) were observed in the transfer curve, and it can be related to the built-in field arising from carrier trapping. Thirdly, the properties of the ion gel are critical for efficient electrostatic gating of the synaptic device. Figure R6 below shows the frequency-dependent specific capacitance of the ion gel film, exhibiting a large specific capacitance of ~1.5 μF cm⁻² at a frequency of 1 kHz. This large capacitance arises from the formation of electrical double layers (EDL) at the interface. Figure R6 below also reveals that the capacitance of the ion gel film is strongly dependent on frequency (up to 10 kHz), since the

mobilities of the ions (in our case the ions are mostly protons with high mobility) limit the polarization response time. [Nat. Mater., 7, 900-906, 2008] The estimated formation time of the double-layer capacitances for the ion-gel electrolyte is in the order of 10 μ s, which allows synaptic response under stimuli of voltage spikes close to 100 kHz. The large specific capacitance of the ion gel and the rapid formation of the EDL layer attributed to the high mobility of the ion can ensure effective electrostatic gating in the synaptic transistor. Besides, the ion gel exhibits high mechanical flexibility (>1000 bending times), good ambient stability (storage life >3 months), and large resistance (>5 M Ω), making it beneficial for flexible electronics applications.

To address the reviewer comment on the stability and repeatability of our strategy of adjusting the sustaining spike's amplitude across multiple experiments, we conduct further analysis. In our work, this strategy aiming at mitigating the device-to-device variations is implemented by finely tuning the amplitude of the sustaining spikes (minimum step size in our setup is 0.05 V) to ensure similar retention behavior among different devices. This strategy has been applied to 12 different devices, and the corresponding retention behaviors were evaluated at $t = 20$ s. The results in Figure R7 below demonstrate that this strategy ensures close performance in retention behavior (variation $\sim 2.6\%$) among different devices in multiple experiments. Additionally, the synaptic response of the device exhibits high stability and repeatability at different spike cycles (variation $\sim 1.1\%$ after 400 cycles), as shown in Figure R8(a) below. Furthermore, the device also demonstrates stability against mechanical bending and long-term storage, as presented in Figure R8(b)-(d) below. Therefore, this strategy of adjusting sustaining spike's amplitude can ensure good stability and repeatability across multiple experiments or across different devices.

To address the reviewer comment on how the semiconducting material and ion gel contribute to the synaptic and memory properties, we provide further clarification. Firstly, regarding the mixed-dimensional materials in our device, the MoS₂ nanoflakes (2D nanomaterial) adsorbed on the metal-oxide thin film (3D film) may serve as charge trapping sites during the electrostatic gating and conductance modulation under the stimuli of voltage spikes. Kelvin-probe force microscopy (KPFM) mapping in Figure R9 below reveals surface potential changes in the region of nanoflakes, which can be attributed to carrier trapping. Moreover, in comparison with the control device fabricated with metal-oxide film alone, the device adsorbed with 2D nanoflakes exhibit improvements in both transistor and synaptic characteristics. Specifically, enhancements of device performance are observed in the on/off ratio (increased from 7.5 to 84.5), the off-state current (suppressed from 3.05 to 0.15 μ A), and the linearity of synaptic behavior (under the stimuli of spike trains). These enhancements facilitate the production of synaptic current with large response and linear potentiation, which is beneficial for achieving synaptic plasticity, such as spike-number dependent plasticity (SNDP). Secondly, regarding the ion gel in our device, the alginate-based ion gel has much smaller ion radius (proton), compared to other electrolyte ion gels containing larger ions such as EMIM⁺ and Li⁺. The high mobility of the ions can ensure ion migration even under high-frequency stimulation, enabling synaptic response of the device under fast spikes. Additionally, the ion gel layer exhibits a large specific capacitance ($\sim 1.5 \mu\text{F cm}^{-2}$ at 1 kHz) attributed to the formation of an electrical double layer (EDL) at the interface, and thus it ensures effective electrostatic gating, which is critical for realizing the synaptic response of the device under voltage spikes at the gate. Moreover, the capacitance of alginate ion gel exhibit frequency-sensitive behavior (up to 100 kHz), enabling spike-frequency dependent plasticity (SFDP) of the synaptic device. The alginate-based ion gel also demonstrates advantages in long-term storage and biocompatibility, making it suitable for wearable electronics. The alginate ion gel and the mixed-dimension semiconductor

materials are compatible with flexible substrates due to their solution-processable fabrication, advantageous for implementing neuromorphic systems towards applications in wearable devices and flexible electronics.

In response to the reviewer’s comment on how the unique features (synaptic and memory properties as well as synaptic and neuronal functions) enhance the neuromorphic capabilities of the device, we add further explanations. Firstly, the synaptic properties of our device, including spike-number dependent plasticity (SNDP), linear potentiation, and cycle-to-cycle stability under repeated stimuli, ensure a stable, repeatable, linear response to voltage spikes. These synaptic characteristics enable the neuromorphic recognition of sensory information encoded as spike trains. Secondly, the frequency-sensitive large capacitance of the ion gel and the high mobility of the ion contribute to the effective electrostatic gating and frequency-dependent spike response in our device, thus enabling neuromorphic capabilities including spike-frequency dependent plasticity (SFDP). Thirdly, the memory properties of the device guarantee the storage of sensory information with controlled sensory memory. This feature enables history-dependent learning rules such as Bienenstock–Cooper–Munro (BCM) and establishes the foundation for neuromorphic processing with sensory memory. Additionally, the strategy of sustaining spikes developed for the synaptic device allows the manual regulation of sensory memory, offering the adaptability for task-specific neuromorphic processing with varying memory requirements. Forth, the neuronal functions and the neuronal architecture of the device, such as multisensory integration (multiple sensory spikes are forwarded to the dual gates of the device as multiple inputs), spatiotemporal recognition (the device is sensitive to the temporal congruence of pairwise spikes obtained from different sensors), and the “labeled-line” model of receptor-neuron pathway (sensor and device are connected based on the type of sensory spike), permit the processing of sensory information encoded as spatiotemporal spikes in a parallel, pulse-driven, multimodal manner, thus enabling the neuromorphic perception of tactile and magneto sensory stimuli.

Figure R1. Schematic illustration of the operating mechanism of the device (this figure also appears in Figure S11 in Supplementary Information).

Figure R2. SEM image of the uniform metal-oxide semiconductor film showing its surface morphology.

Figure R3. TEM image of the purified MoS₂ nanoflake showing its crystalline structure (this figure also appears in Figure S7(c) in Supplementary Information).

Figure R4. SEM image of the MoS₂ nanoflakes distributed at the surface of semiconducting material without aggregation.

Figure R5. Performance comparison of the control device and the nanoflake-adsorbed device (this figure also appears in Figure S10 in Supplementary Information). (a) Transfer curves of the two devices in linear scale. (b) Transfer curves of the two devices in log scale. (c) Spike-number dependent plasticity of the two devices under voltage spikes at 20 Hz. (d) Spike-number dependent plasticity of the two devices under voltage spikes at 100 Hz.

Figure R6. Frequency-dependent specific capacitance of the ion gel film (this figure also appears in Figure S12 in Supplementary Information).

Figure R7. The strategy of adjusting sustaining spikes across multiple devices (this figure also appears in Figure S14 in Supplementary Information). (a) Finely regulation of the memory retention behavior of the device. (b) Memory retention behaviors ($t = 20$ s) of 12 different devices after implementing the strategy across them.

Figure R8. Stability and repeatability of the device against cycling, bending, and long-term storage (this figure also appears in Figure S13 in Supplementary Information). (a) Cycling test

of the synaptic device. (b) Photograph of the synaptic device fabricated as 5 by 5 array on a flexible substrate. (c) Bending test of the synaptic device. (d) Storage test of the synaptic device.

Figure R9. Surface potential mapping of the semiconductor channel in the device (this figure also appears in Figure S9 in Supplementary Information). (a) Topology mapping acquired by AFM showing the MoS₂ nanoflakes distributed on the surface of a uniform metal-oxide film. (b) Surface potential mapping acquired by KPFM revealing the surface potential change on the locations of the nanoflakes.

The reviewer may refer to the following revised content in the supplementary information:

- Supplementary Note 7. Evaluation and improvement of device-to-device variations.

Comment 2 from Reviewer #3:

In response to comment 3, the authors have provided clarifications on the selection of FA and SA spikes in various classification tasks. Based on the explanations provided by the authors, it seems that the adaptive selection of spikes (FA and SA) is crucial for achieving high-performance artificial antennal sensory systems. Could the authors elaborate on whether the suggested neuromorphic system can dynamically adjust its spike selection strategy in real-time based on changes in the environment or task requirements? Additionally, does the proposed system possess any learning or adaptive capabilities in terms of adjusting its spike selection strategy over time?

Author Response:

We would like to thank the reviewer for this insightful suggestion.

The adaptive selection of sensory spikes, including fast adaptation (FA) and slowly adaptation (SA), plays a pivotal role in achieving high performance in artificial antennal sensory systems. Indeed, the precise choice of FA and SA spike is essential for attaining high levels of efficiency and recognition accuracy.

In the previous version of the response letter, the selection of FA and SA spikes was manually chosen based on the changing behavior of the sensory stimuli. Based on the reviewer's suggestion, we propose a

feasible method for the dynamic adjustment of the spike selection strategy. This involves evaluating the mean firing rates of sensory spikes during a tactile perception event. Specifically, the microcontroller in the spike-encoding circuit can be programmed to compare the mean firing rates of FA and SA spikes. For instance, in chess recognition task, computing the mean firing rates of sensory spikes over the duration of a tactile event reveals a significantly higher firing rate for SA spike (as shown in the Figure R10 below). Consequently, the microcontroller can dynamically select the sensory signal with higher firing rate for further synaptic processing, enabling more accurate recognition. By incorporating this function into the microcontroller, the neuromorphic system can achieve the dynamic adjustment of the spike selection strategy. This dynamic adjustment is performed in an event-based manner to select reliable sensory spike based on mean firing rate, and it can potentially address the changes in the environments (e.g. alterations in the frequency or amplitude of tactile stimuli) or task requirements (e.g. profile recognition or surface recognition).

Figure R10. Encoded sensory spikes (SA and FA spikes) for chess profile classification (this figure also appears in Figure S15 in Supplementary Information).

To realize the learning or adaptive capabilities of our neuromorphic system in terms of adjusting the spike selection strategy, we propose an approach inspired by the multilayer perceptron (MLP) model with backpropagation functions, as depicted in Figure R11 below. This approach involves the backward transmission of information from a later layer to a previous layer, thus reducing recognition error and enabling the learning function. Note that the original architecture of our neuromorphic system transmits information from the flexible sensor to the synaptic device in one direction, similar to the notion of feedforward neural network (FNN) in machine learning. To achieve the perception learning function, our system can be further integrated with a feedback circuit that forwards the device output to the microcontroller in the spike-encoding circuit. This integration forms feedback loops and allows the bi-directional flow of sensory information, similar to the notion of recurrent neural network (RNN) in machine learning. The microcontroller can be programmed to evaluate the outputs from the SA and FA devices (e.g., by comparing the intensity of the synaptic current at the end of a perception event), and then select the suitable sensory spike (SA or FA spike) for sensory processing and recognition.

From a biological perspective, the proposed design, which integrates feedback loops in our neuromorphic system, essentially emulates the organization of a biological brain with feedback mechanism. The selection of sensory spike signals is analogous to the brain function of perceptual weighting, where the biased assignment of perceptual weights to specific sensory stimuli depends on the reliability and intensity of sensory inputs.

Figure R11. Multilayer perceptron model with backpropagation. Feedback of information is involved in this model.

Above all, we propose a method for dynamic adjustment of the spike selection strategy by comparing the mean firing rates of different sensory spikes (FA and SA spikes) in the microcontroller. As an extension and future endeavor, we also propose an approach that emulates the learning functions in a multi-layer perceptron model and incorporates feedback loops to compare the outputs of FA and SA devices, aiming at achieving the learning capabilities for selecting more suitable sensory spike.

The reviewer may refer to the following revised content in the supplementary information:

- **Supplementary Note 15. Fast-adapting (FA) and slowly-adapting (SA) spikes.**

The reviewer may also refer to the following revised content in the manuscript:

- **Experimental: “Additionally, dynamic selection of the sensory spikes suitable for sensory processing can be achieved by comparing the mean firing rates of the FA and SA spikes in the microcontroller.”**

Comment 3 from Reviewer #3:

Regarding the response to comment 4, the authors noted that the optimal scanning speed for profile detection and recognition (classification) may differ. It would be helpful for the authors to elaborate on why there is a variation in the optimal scanning speed for different tactile perception tasks, such as profile detection and surface pattern recognition (classification). Since profile detection should logically precede

recognition, it seems there should be no difference in optimal scanning speed between the two tasks. Also, how does the varying speed align with human tactile exploration and the specific requirements of each task?

Author Response:

We appreciate the reviewer’s comment.

Firstly, we would like to clarify that the first sentence of the reviewer’s comment, “the optimal scanning speed for profile detection and recognition (classification) may differ”, did not precisely convey the intended meaning in our first version of the response letter. The key idea we aim to convey is that the optimal scanning speeds for profile classification and surface classification are distinct. To align with this clarification, we have replaced “profile detection” with “profile classification” in response to this reviewer comment, because essentially the profiles of chess pieces are not only detected using the flexible sensor but also classified using the synaptic device during the tactile perception experiment, as shown in Figure 4(c)-(e) in the manuscript. Correspondingly, we have revised both the manuscript and supplementary information to describe the perception tasks of “profile classification” and “surface classification” in a clear and understandable manner.

Secondly, we attend to address the reviewer’s comment about elaboration of the variations in optimal scanning speed for different tactile perception tasks. In profile classification task as illustrated in Figure R12(a) below, the shape of the chess piece needs to be identified, given that the chess piece has smooth surface but large irregular contour. The lateral scanning process mainly induces the significant deflection (large deflection >3 mm) of the flexible sensor, resulting in the prominent response in the SA sensory spikes. A relatively slow scanning speed can potentially ensure the gradual deformation of the flexible sensor and the effective accumulation of synaptic current from the SA device. In surface/texture classification task as shown in Figure R12(b) below, the morphology of the surface needs to be perceived, given that the sample surface is flat but has small periodic stripes or textures. The lateral scanning operation mainly induces repeated vibrations of the flexible sensor (small deflection <0.5 mm), leading to the intense response in the FA sensory spikes. A relatively fast scanning speed can ensure the intense vibration of the sensor tip during scanning across the pattern or texture, resulting in large output from the FA device.

Figure R12. Experimental setups for different tactile perception tasks (this figure also appears in Figure S24(a)-(b) in Supplementary Information and Figure 4(b) in Manuscript). (a) Profile classification task. (b) Surface/texture classification task.

Finally, we further analyze the human tactile exploration experiment. Two video clips were provided (uploaded to FigShare repository with DOI: 10.6084/m9.figshare.24980661) to demonstrate the finger motions of a volunteer during “blind” tactile explorations of a chess piece and a sample surface. The observed scanning speeds of the volunteer’s finger for perceiving surfaces and shapes are quite different, as shown in Figure R13 below. The preferred scanning speed of the volunteer’s finger was slower when identifying chess shapes, in contrast to the faster scanning speed employed when identifying surface patterns. This discrepancy suggests that human employs distinct scanning speeds of finger motion during surface recognition task and profile recognition task to achieve optimal perceptual performance. In our original manuscript, we have mentioned that all the participants were instructed to perceive the chess piece using one finger at their preferred speed. For our neuromorphic system, the optimal scanning speed of the sensor also varies with perception tasks. The variation is attributed to the distinct goals of surface recognition and profile recognition, aimed at identifying surface texture and shape morphology, respectively. Texture information is obtained through the vibration of the flexible sensor, while shape information is acquired by the deformation of the flexible sensor. To achieve highest recognition accuracy, the optimal scanning speed for chess profile recognition is approximately 4 mm s^{-1} , while the optimal scanning speed for surface recognition is approximately 6 mm s^{-1} . Therefore, the variations in optimal scanning speeds of the sensor among different perception tasks briefly match with the observations in humans, who employed various preferred speeds of finger motions for distinct tactile explorations. Correspondingly, we have revised the supplementary information to include the analysis on the varied optimal scanning speed for different perception tasks.

Figure R13. Human tactile experiments demonstrating the varied scanning speed of the finger in different perception tasks. (a) Tactile exploration in surface recognition task. (b) Tactile exploration in chess recognition task.

The reviewer may refer to the following revised content in the supplementary information:

- **Supplementary Note 10. Influence of scanning speed on tactile recognition.**

The reviewer may also refer to the following revised content in the manuscript:

- The term “**profile detection**” is replaced by “**profile classification**”.

REVIEWERS' COMMENTS

Reviewer #3 (Remarks to the Author):

The authors well revised their manuscript according to reviewers' comments. It is acceptable for publication in this journal.

Response to Reviewers' Comments
(Manuscript ID: NCOMMS-23-49599-B)

Reviewer #3

Comments:

The authors well revised their manuscript according to reviewers' comments. It is acceptable for publication in this journal.

Author Response:

We sincerely appreciate the reviewer's recommendation of our work. We also extend our gratitude for the time and effort the reviewer dedicated to the review.